# BREAKING BARRIERS: DO REINFORCEMENT POST TRAINING GAINS TRANSFER TO UNSEEN DOMAINS?

**Chuxuan Hu[1], Yuxuan Zhu[1], Antony Kellermann, Caleb Biddulph,**
**Suppakit Waiwitlikhit, Jason Benn, Daniel Kang[1]**
[1]Siebel School of Computing and Data Science, University of Illinois at Urbana-Champaign
`{chuxuan3, yxx404, ddkang}@illinois.edu`

## ABSTRACT

Reinforcement post training (RPT) has recently shown promise in improving the reasoning abilities of large language models (LLMs). However, it remains unclear how well these improvements generalize to new domains, as prior work evaluates RPT models on data from the same domains used for post-training. To understand the generalizability of RPT, we conduct two studies with specific focus on Reinforcement Learning with Verifiable Rewards (RLVR). (1) Observational: we compare a wide range of open-weight RPT models against their corresponding base models across multiple domains, including both seen and unseen domains in their fine-tuning data. (2) Interventional: we fine-tune LLMs with RPT on single domains and evaluate their performance across multiple domains. Both studies converge on the same conclusion that, although RPT brings substantial gains on tasks similar to the fine-tuning data, the gains generalize inconsistently and can vanish on domains with different reasoning patterns.

## 1 INTRODUCTION

Large language models (LLMs) demonstrate strong capabilities across diverse reasoning settings. In mathematics and quantitative reasoning, recent models achieve near-expert performance on benchmarks including GSM8K, MATH, and AIME (Cobbe et al., 2021; Lightman et al., 2023; Jia, 2024; Tunstall & Jia, 2024). In code generation and program synthesis, LLMs similarly show substantial progress across challenging evaluations (Austin et al., 2021; Chen et al., 2021; Jain et al., 2024; Shi et al., 2024; Zhuo et al., 2025; MatrixStudio, 2024; Aider-AI, 2025). Beyond these structured domains, LLMs also perform well on knowledge-intensive reasoning spanning legal, financial, and biomedical tasks (Guha et al., 2023; Zhang et al.; Jin et al., 2019; Griot et al., 2025). A growing body of work shows that reinforcement post-training (RPT) can yield large performance gains on these tasks. Recent RPT-enhanced models achieve dramatic improvements in competition-level mathematics and coding benchmarks, in some cases matching or exceeding top human competitors (Shao et al., 2024; Luo et al., 2025a;b; Su et al., 2025b; Zhao et al., 2025a; Yuan et al., 2024; Cui et al., 2025; Xie et al., 2025; Team, 2025a; He et al., 2025; Liu et al., 2025; Google DeepMind, 2025a; xAI, 2025; Granite Embedding Team, 2024; OpenAI et al., 2024; Anthropic, 2025). This raises an important question: does RPT provide generalizable improvements, as broadly as those achieved through pretraining?

Existing evaluation frameworks and RPT setups provide limited evidence to answer this question. To address it systematically, we design a two-stage investigation pipeline (Figure 1a), with a specific focus on Reinforcement Learning with Verifiable Rewards (RLVR), the dominant instantiation of RPT for verifiable reasoning tasks.

First, prior work evaluates RPT models within their post-training domains (Luo et al., 2025b;a). To overcome this limitation, we conduct an *observational study* in which we evaluate 18 recent open-weight RPT models with publicly disclosed post-training data alongside their corresponding base models across a wide range of domains, including legal, financial, and medical benchmarks, spanning their seen and unseen domains. This study is designed to provide an initial view into the generalizability of RPT.

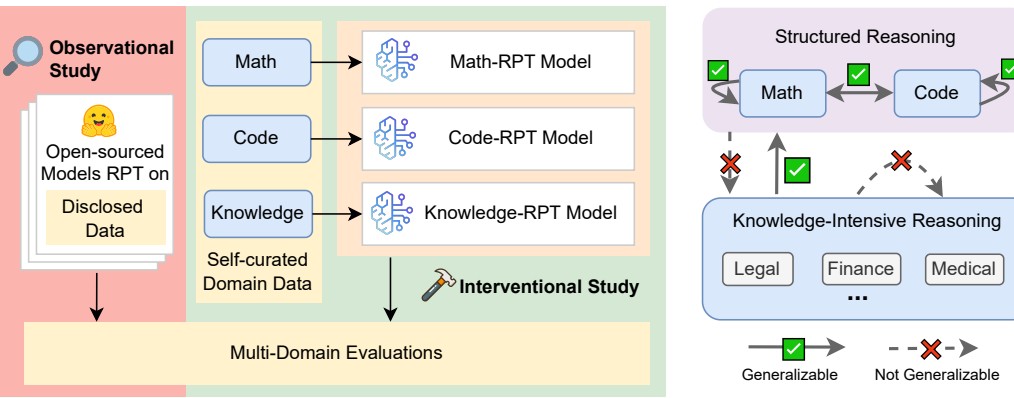

(a) Overview of our two-stage investigation pipeline.    (b) Summary of RPT generalizability.

Figure 1: The method (a) and key findings (b) of our work. Through a unified multi-domain evaluation framework combining observational and interventional studies, we find that RPT exhibits limited generalizability across domains.

Additionally, we notice that these RPT models, as a representative selection of existing open-weight models such as DeepSeek R1 (DeepSeek-AI et al., 2025), are post-trained on mixed domain data. The presence of such confounding factors makes it difficult to isolate and interpret the generalizability of RPT at a finer granularity. To strengthen our findings, we conduct an *interventional study* in which we post-train LLMs via RPT on math, coding, and knowledge-intensive reasoning data and evaluate their performance on both in-domain and out-of-domain tasks. We illustrate our methodology in detail in Section 3.

As we summarize in Figure 1b, our findings show that gains from RPT shows limited generalizabilty. Specifically, RPT gains on domains involving structured reasoning patterns (e.g., math, code) generalize well within and across structured domains, but fail to generalize to unstructured domains. In contrast, gains from RPT on unstructured domains (e.g., legal, finance) do not generalize well within unstructured domains, but show transferability to structured domains. We analyze these results comprehensively in Section 4.

Our findings suggest that RPT models are most effective when the target task shares similar reasoning patterns with the RPT data. Consequently, while RPT remains a powerful method for improving LLMs' performance, its benefits are largely limited to the domains represented in the post-training data and do not generalize to a wide range of new, unseen domains.

## 2  BACKGROUND

In this section, we introduce the motivation of our study. We begin by demonstrating the strong performance of LLMs post-trained via RPT across various reasoning tasks, particularly in mathematics and coding. We then discuss the limitations of existing work in understanding the mechanisms and boundaries of RPT. We close by introducing the data-domain taxonomy that grounds our study.

**RPT models demonstrate promising performance across a wide range of tasks.**  RPT models have achieved remarkable improvements on complex reasoning benchmarks. For example, Gemini 3 Pro (Google DeepMind, 2025b) achieves 100% accuracy on AIME 2025 (OpenCompass Community, 2025), a math competition benchmark. Claude Opus 4.5 (Anthropic, 2025) reaches 81% accuracy on SWE-bench Verified (Jimenez et al., 2024), a coding benchmark, while GPT-5.2-Pro (OpenAI, 2025) reaches around 93.2% accuracy on GPQA Diamond (Rein et al., 2023), a benchmark for graduate-level scientific reasoning.

**Out-of-domain generalizability of RPT models remains understudied.**   Despite impressive results, recent work has examined the limitations of RPT models and the opaque nature of their underlying reasoning capabilities (Team, 2025b; Yue et al., 2025; Ma et al., 2025; Ye et al., 2025).

In particular, there is growing interest in the role of RPT data, especially the extent to which RPT algorithms rely on large, diverse corpora to achieve generalization (Zhao et al., 2025a).

While RPT models benefit from training on diverse, mixed-domain data, this diversity makes it difficult to directly assess their generalizability to unseen domains. As a result, prior work evaluates RPT models on tasks within the same domains as their training data (Team, 2025b; Yue et al., 2025; Ma et al., 2025; Ye et al., 2025). However, many reasoning tasks remain underrepresented or entirely absent from existing training corpora. Exploring the generalizability of LLMs trained with RPT to such tasks is therefore essential for identifying the boundaries of their applicability in real-world, complex scenarios.

**Understanding RPT generalizability requires a systematic view of reasoning domains.** Following the task suites defined in prior work (Su et al., 2025a), we focus on three major *domains* of interest: code, math, and knowledge-intensive reasoning (Figure 1b). These domains are chosen to capture a broad spectrum of reasoning challenges commonly seen in language model evaluations. The knowledge-intensive reasoning domain can be further divided into application-specific subdomains such as legal, finance, medical, etc. We not only examine performance across the high-level domains, but also evaluate how reasoning patterns and generalization behaviors vary across subdomains.

Among the three, we consider math and code domain tasks to follow structured reasoning patterns, where solutions follow deterministic logical steps and require precise syntax and formal semantics (Su et al., 2025a). In contrast, tasks within the knowledge-intensive reasoning domain require more flexible and context-sensitive reasoning, referred to here as *unstructured* reasoning. We define unstructured reasoning as problem-solving processes that do not adhere to a fixed sequence of logical operations and lack a well-defined intermediate representation or symbolic grounding. Such tasks demand broader world knowledge, interpretive judgment, and the ability to handle ambiguity or incomplete information. For instance, legal and financial question answering may involve interpreting lengthy documents, extracting relevant information from loosely connected statements, or evaluating conflicting evidence. We quantify the rationale of such divisions in Appendix B.

## 3 STUDY DESIGN

We present our study design that aims to investigate the generalizability of RPT. We propose the following research questions (RQs) that have not been systematically examined in prior work.

- **(RQ1) Cross-domain generalization.** To what extent do the capabilities acquired through RPT transfer to tasks from domains not included in the training data?

- **(RQ2) Role of reasoning structure.** How does the structure of reasoning required by a task affect generalization? Do skills learned from highly structured domains (e.g., mathematics, code generation) transfer to less-structured domains (e.g., medical or legal reasoning), and vice versa?

- **(RQ3) Intra-domain generalization.** How effectively do RPT gains generalize across subdomains within the same domain?

- **(RQ4) Stability of generalization.** Is the generalizability of RPT consistent across different hyperparameters, such as model size, training algorithm, and the number of RPT steps?

To address these research questions, we design a two-stage pipeline. First, we perform an observational study by evaluating 18 RPT models, each compared against its corresponding base model across a diverse set of benchmarks, spanning their seen and unseen domains. Because existing RPT models are typically trained with different configurations (e.g., different RL algorithms and hyperparameters) on multi-domain data, it is challenging to isolate the effect of RPT itself from the advantages brought by specific configuration or data.

To mitigate confounding factors, we further conduct an interventional study, where we post-trained three RPT models from the same base model with the same configuration, each on a disjoint single-domain dataset. We then evaluate these trained models using the same benchmarks as in the observational analysis. In the rest of this section, we describe our evaluation settings and experiment setups for both studies.

## 3.1 EVALUATION SETTINGS

**Benchmarks.** For evaluation, we use 16 popular benchmarks, covering a wide range of domains and difficulty levels. We categorize these benchmarks into the following three representative domains:

- *Math*: For easier questions, we use GSM8K (Cobbe et al., 2021) and MATH-500 (Lightman et al., 2023), while for more challenging problems, we select AIME 2024 (Jia, 2024) and AMC 2023 (Tunstall & Jia, 2024).

- *Code*: We use easy coding problems, including MBPP (Austin et al., 2021) and HumanEval (Chen et al., 2021), and relatively challenging problems, including BigCodeBench (Zhuo et al., 2025), LiveCodeBench (Jain et al., 2024), USACO (Shi et al., 2024), and Codeforces (MatrixStudio, 2024). To test programming language generalization, we also include the multi-language benchmark Polyglot (Aider-AI, 2025).

- *Knowledge-intensive reasoning*: We use high-quality benchmarks that are not mathematics nor programming problems for knowledge-intensive reasoning, including PubMedQA (Jin et al., 2019) and MedQA (Griot et al., 2025) for medical reasoning, TabFact (Chen et al., 2019) for fact verification, LegalBench (Guha et al., 2023) for legal reasoning, and FinBench (Zhang et al.) for financial problem solving.

**Generation Configurations.** For all benchmarks, we use a consistent set of generation hyperparameters across all models. The maximum response length is set to $\min\{16192, C\}$, where $C$ denotes the model's context window. For each model, we run the small benchmarks (i.e., AMC 2023 and AIME 2024) 16 times, while executing all other benchmarks once. For prompting, we apply each model's default chat template and system prompt.

**Evaluation Metrics.** To assess whether RPT improves the accuracy performance within or across domains, we report the aggregated accuracy improvement $\Delta_{i,j}^{(\mathcal{D})}$ of an RPT model $i$ over its base model $j$ for a given domain $\mathcal{D}$:

$$\Delta_{i,j}^{(\mathcal{D})} = \frac{\sum_{t\in\mathcal{D}} N_t R_t (A_{i,t} - A_{j,t})}{\sum_{t\in\mathcal{D}} N_t R_t},$$

where $N_t$ is the number of problems in $t$, $R_t$ is the number of repetitions we executed for $t$, $A_{i,t}$ is the accuracy of model $i$ on benchmark $t$, and $A_{j,t}$ is the accuracy of model $j$ on benchmark $t$.

In addition, to ensure statistical significance in our findings, we applied the Cochran–Mantel–Haenszel (CMH) test (Agresti, 2013), a statistical test for analyzing stratified categorical data. We treat each benchmark as an independent stratum—that is, a random sample of distinct downstream tasks. Given an RPT model $i$, a base model $j$, and a domain $\mathcal{D}$ of benchmarks, we calculate the common odds ratio estimate ($\theta_{i,j,\mathcal{D}}$) that estimates the correlation between the RPT process and the accuracy improvement on $\mathcal{D}$:

$$\hat{\theta}_{i,j}^{(\mathcal{D})} = \frac{\sum_{t\in\mathcal{D}} N_t R_t A_{i,t}(1 - A_{j,t})}{\sum_{t\in\mathcal{D}} N_t R_t A_{j,t}(1 - A_{i,t})}$$

An odds ratio greater than 1 indicates improvement from RPT; a value less than 1 indicates a decrease in accuracy from RPT. We evaluate the statistical significance under the null hypothesis $H_0 : \theta_{i,j}^{(\mathcal{D})} = 1$ against the alternative hypothesis $H_1 : \theta_{i,j}^{(\mathcal{D})} \neq 1$, using the standard CHM test statistics,

$$\xi = \frac{\left(\sum_{t\in\mathcal{D}} N_t R_t (A_{i,t} - A_{j,t})\right)^2}{\sum_{t\in\mathcal{D}} N_t^2 R_t^2 A_{i,t} A_{j,t}(1 - A_{i,t})(1 - A_{j,t})(2N_t - 1)^{-1}}$$

which follows a chi-squared distribution asymptotically with 1 degree of freedom. For all reported values, an asterisk (*) denotes statistical significance at $p < 0.05$.

## 3.2 EXPERIMENTAL SETTINGS

**Observational Study.** To ensure a comprehensive and representative evaluation of RPT model generalizability, we adopt a systematic approach to selecting models for our observational study:

Table 1: Selected RFT models for observational analysis. The `RFT Domain(s)` refers to the domain(s) covered in the RFT training data.

| *(Model ID)* **RPT Model** | **Base Model** | **RPT Domain(s)** |
|---|---|---|
| *(1)* DeepScaleR-1.5B-Preview (Luo et al., 2025b) | DeepSeek-R1-Distill-Qwen-1.5B (DeepSeek-AI et al., 2025) | Math |
| *(2)* DeepCoder-1.5B-Preview (Luo et al., 2025a) | DeepSeek-R1-Distill-Qwen-1.5B (DeepSeek-AI et al., 2025) | Code |
| *(3)* Skywork-o1-Open-Llama-3.1-8B (o1 Team, 2024) | Llama-3.1-8B-Instruct (Meta AI, 2024b) | Code, Math |
| *(4)* Eurus-2-7B-PRIME (Cui et al., 2025; Yuan et al., 2024) | Eurus-2-7B-SFT (Cui et al., 2025; Yuan et al., 2024) | Code, Math |
| *(5)* Absolute_Zero_Reasoner-Coder-3b (Zhao et al., 2025a) | Qwen2.5-Coder-3B (Hui et al., 2024; Yang et al., 2024) | Code |
| *(6)* Absolute_Zero_Reasoner-Coder-7b (Zhao et al., 2025a) | Qwen2.5-Coder-7B (Hui et al., 2024; Yang et al., 2024) | Code |
| *(7)* ZR1-1.5B (Zyphra, 2024) | DeepSeek-R1-Distill-Qwen-1.5B (DeepSeek-AI et al., 2025) | Code, Math |
| *(8)* Llama-3.1-Nemotron-Nano-8B-v1 (Bercovich et al., 2025) | Llama-3.1-8B-Instruct (Meta AI, 2024b) | Instruction Following |
| *(9)* Thespis-Llama-3.1-8B (Locutusque, 2024) | Meta-Llama-3.1-8B-Instruct-abliterated (mlabonne, 2024) | Chat |
| *(10)* STILL-3-1.5B-preview (Team, 2025d; Jiang et al., 2024; Min et al., 2024) | DeepSeek-R1-Distill-Qwen-1.5B (DeepSeek-AI et al., 2025) | Math |
| *(11)* Arcee-Maestro-7B-Preview (AI, 2024) | DeepSeek-R1-Distill-Qwen-7B (DeepSeek-AI et al., 2025) | Code, Math |
| *(12)* Fino1-8B (Qian et al., 2025) | Llama-3.1-8B-Instruct (Meta AI, 2024b) | Finance |
| *(13)* OREAL-7B (Lyu et al., 2025) | OREAL-7B-SFT (Lyu et al., 2025) | Math |
| *(14)* Open-RS3 (Dang & Ngo, 2025) | DeepSeek-R1-Distill-Qwen-1.5B (DeepSeek-AI et al., 2025) | Math |
| *(15)* DeepCoder-14B-Preview (Luo et al., 2025a) | DeepSeek-R1-Distill-Qwen-14B (DeepSeek-AI et al., 2025) | Code |
| *(16)* OREAL-32L (Lyu et al., 2025) | OREAL-32B-SFT (Lyu et al., 2025) | Math |
| *(17)* Fin-o1-14B (Qian et al., 2025) | Qwen3-14B (Team, 2025c) | Finance |
| *(18)* Absolute_Zero_Reasoner-Coder-14b (Zhao et al., 2025a) | Qwen2.5-Coder-14B (Hui et al., 2024; Yang et al., 2024) | Code |

- *Stage 1.* We collect the 466 models from Hugging Face applying the following filtering criteria: as of April 23rd, 2025: (1) the model supports `Text Generation` tasks, (2) its model card description contains the keyword `reasoning`, `chain-of-thought`, and/or `chain of thought` and (3) the model has received at least 10 likes.

- *Stage 2.* We use `o4-mini` (OpenAI, 2025) to prefilter models potentially trained with RPT, based on their model card descriptions. This automatic filtering is followed by manual verification, resulting in 31 models that we confirm to be RPT models.

- *Stage 3.* From the 31 RPT models, we manually select 12 that meet the following criteria: (1) the RPT datasets are publicly disclosed, (2) the model sizes range from 1.5B to 8B parameters, and (3) the base models are not purely pretrained models, ensuring they can generate coherent responses and follow basic instructions for evaluating reasoning capabilities.

We evaluate `Absolute_Zero_Reasoner-Coder-3B`, post-trained with limited RPT data but demonstrating strong performance on math and code reasoning tasks. We view it as a representative case for examining RPT generalizability. We also evaluate the variants of all selected RPT models with more parameters, whenever such variants are available.

We finalize our selection of 18 RPT models, with the details, including base models and RPT domains, presented in Table 1. For each RPT model and its corresponding base model, we compare performance across 16 benchmarks.

**Interventional Study.** To isolate the effect of RPT from other training configurations, including datasets, algorithms, and hyperparameters, we trained three RPT models based on `DeepSeek-R1-Distill-Qwen-1.5B` (DeepSeek-AI et al., 2025) on three disjoint datasets, math, code, and knowledge-intensive reasoning, respectively. We curated training datasets based on existing datasets that led to performant RPT models and cleaned them to ensure that the training datasets do not overlap with our evaluation sets. We built the following datasets:

- *Math*: we uniformly sampled 40,000 problems from a combination of the math split of Eurus-2-RL (Cui et al., 2025), which originates from the NuminaMath-CoT dataset (LI et al., 2024).

- *Code*: we uniformly sampled 40,000 deduplicated problems from a combination of KodCode (Xu et al., 2025), DeepCoder-Preview (Luo et al., 2025a), Apps (Hendrycks et al., 2021), TACO (Li et al., 2023), and the code split of Eurus-2-RL (Cui et al., 2025).

- *Knowledge-intensive Reasoning*: we selected 40,000 high-quality, non-math, and non-code data from the multi-subject RPT dataset (Su et al., 2025b). To achieve that, we applied o3-mini (OpenAI, 2025) to exclude math-related, code-related, or fact-recall questions.

We applied consistent settings for all three RPT training processes. In terms of the RL algorithm, we applied Group Relative Policy Optimization (GRPO) with the same setting as DeepCoder (Luo et al., 2025a) , with the reward definitions and dynamics included in Appendix D. In terms of hyperparameters, we trained each dataset for one epoch with a batch size of 64 and a context length

of 8,192. To stabilize the training process, we used a learning rate of $10^{-6}$ and an entropy coefficient of 0. We post-trained the models on 8 80GB H100 GPUs.

Furthermore, we conducted three additional experiments under alternative settings, to validate our conclusions about the generalizability of RPT,

1. We trained with DAPO (Yu et al., 2025), a state-of-the-art RL algorithm.
2. We extended the training process to 2 epochs and evaluated intermediate checkpoints.
3. We trained with `Llama-3.2-3B-Instruct` (Meta AI, 2024a) as the base model.

## 4 FINDINGS

In this section, we present the findings of our study based on results from observational and interventional studies. We summarize our findings as follows:

- *(RQ1)* RPT does not exhibit generalizability in arbitrary unseen domains (Section 4.1).
- *(RQ2)* RPT demonstrates cross-domain generalizability when reasoning patterns are similar, such as mutual transfer between math and code, but fails to generalize across distinct reasoning patterns, such as from math or code to knowledge-intensive reasoning (Section 4.2).
- *(RQ3)* Intra-domain generalizability of RPT strongly depends on the structural similarity between subdomain tasks (Section 4.3).
- *(RQ4)* The generalizability of RPT is consistent across different hyperparameters (Section 4.4).

### 4.1 RPT GAINS DO NOT GENERALIZE TO ARBITRARY UNSEEN DOMAINS

**Existing RPT models fail to transfer beyond their training domains.** We begin by analyzing our observational study, which evaluates a diverse set of existing RPT models using multi-domain tasks. Specifically, we compare the performance improvements of each model in tasks from the same domain as their training data ($ID$), and tasks that are out-of-domain with their training data ($OOD$). For instance, *(1)* `DeepScaleR-1.5B-Preview` is trained exclusively on math-related data. Therefore, $ID$ tasks for this model include GSM8K, MATH500, AIME 2024, and AMC 2023, while all other tasks (e.g., legal, medical, coding) are $OOD$.

We present the results in Table 2. Across the table, RPT models exhibit considerably higher improvements on *ID* tasks compared to *OOD* tasks, with a 2.87% increase in pass@1 for ID tasks, but a 3.19% decrease for OOD tasks. For example, *(1)* DeepScaleR-1.5B-Preview shows a 5.1% gain in pass@1 on math domain tasks, but only 1.7% in others, representing a $3\times$ drop. This lack of generalizability stems from the RPT algorithm itself rather than simply from overfitting to large-scale training data: notably, a similar trend is observed in *(6)* `Absolute_Zero_Reasoner-Coder-7B`, which was post-trained on a near-zero amount of data. Despite its minimal training data exposure, this model experiences a 23.31% decrease in pass@1 accuracy on unseen domains, while achieving a 30.12% improvement within its RPT domain. We also observe that the generalizability of RPT algorithms is sensitive to the training data, implementation details, and finetuning strategy. For example, although all trained on math data, *(1)* `DeepScaleR-1.5B-Preview` demonstrates improvements in both $ID$ and $OOD$ tasks, whereas *(7)* `ZR1-1.5B` and *(10)* `STILL-3-1.5B-Preview` show statistically significant performance gains in $ID$ tasks as well as statistically significant performance drops on $OOD$ tasks. These findings suggest that the gains from RPT are largely domain-specific: models significantly improve on tasks similar to their training data, but fail to generalize robustly to other unseen domains.

**Single-domain finetuning reinforces evidence of RPT's limited generalizability.** To further dissect the generalizability limitations identified in our observational study, we conduct a more controlled investigation by isolating models post-trained exclusively on single domains. To do so, we analyze our interventional study results, where $ID$ corresponds to the training domain, while $OOD$ include all tasks from the remaining two domains in our evaluation.

As shown in Figure 2, none of the models post-trained on a single domain exhibit statistically significant improvement on $OOD$ tasks. Both the Math-RPT model and the Code-RPT model show

Table 2: Existing Open-sourced RPT models achieve significantly larger accuracy gains $\Delta$ (%) and odds ratios $\hat{\theta}$ on in-domain (*ID*) tasks compared to out-of-domain (*OOD*) tasks.

| Metric | (1) | (2) | (3) | (4) | (5) | (6) | (7) | (8) | (9) | (10) | (11) | (12) | (13) | (14) | (15) | (16) | (17) | (18) | Avg. |
|---|---|---|---|---|---|---|---|---|---|---|---|---|---|---|---|---|---|---|---|
| $\Delta^{(ID)}$ ↑ | 5.40 | 4.61 | 6.96 | 11.62 | -6.27 | 30.12 | 2.82 | 7.07 | -26.01 | 4.31 | -4.27 | -3.84 | -0.41 | -0.59 | -1.87 | 0.61 | -3.60 | 25.01 | **2.87** |
| $\Delta^{(OOD)}$ ↑ | 1.67 | 4.01 | -26.41 | -3.70 | 2.55 | -23.31 | -5.47 | 13.22 | -4.73 | -1.33 | -7.39 | -27.89 | 8.44 | -0.34 | -5.68 | -0.04 | -5.43 | 24.34 | **-3.19** |
| $\Delta^{(ID)} - \Delta^{(OOD)}$ | 3.73 | 0.60 | 33.37 | 15.32 | -8.82 | 53.43 | 8.30 | -6.15 | -2.13 | 5.64 | 3.12 | 24.05 | -8.85 | -0.25 | 3.81 | 0.65 | 1.83 | 0.67 | **6.07** |
| $\hat{\theta}^{(ID)}$ ↑ | 1.36* | 1.45* | 1.59* | 2.37* | 0.52* | 22.47* | 1.22* | 1.34* | 0.34* | 1.28* | 0.72* | 0.83* | 0.97 | 0.97 | 0.87* | 1.06 | 0.85* | 15.58* | **3.10** |
| $\hat{\theta}^{(OOD)}$ ↑ | 1.07* | 1.18* | 0.31* | 0.86* | 1.15* | 0.41* | 0.80* | 1.98* | 0.68* | 0.95* | 0.69* | 0.30* | 1.47* | 0.99 | 0.71* | 1.00 | 0.70* | 8.50* | **1.32** |
| $\hat{\theta}^{(ID)}/\hat{\theta}^{(OOD)}$ | 1.27 | 1.22 | 5.15 | 2.75 | 0.45 | 54.61 | 1.53 | 0.68 | 0.50 | 1.35 | 1.03 | 2.74 | 0.66 | 0.98 | 1.22 | 1.06 | 1.20 | 1.83 | **4.46** |

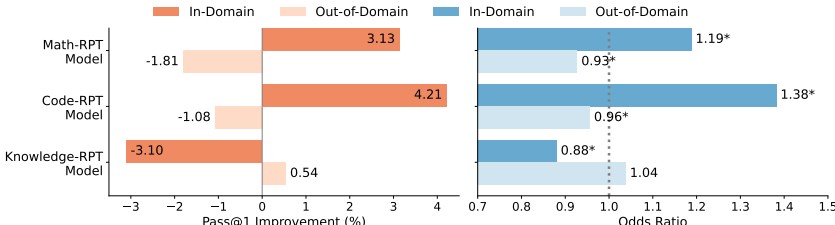

Figure 2: RPT models on single domains show significant pass@1 improvements over base models and higher odds ratios on in-domain tasks, but not on out-of-domain tasks. No single-domain model achieves statistically significant gains in out-of-domain tasks.

performance drops on *OOD* tasks with statistical significance, in contrast to the statistically significant gains they achieve in-domain. The Knowledge-RPT model also demonstrates no statistically significant gains on its *OOD* tasks.

## 4.2 RPT Gains Generalize Across Domains with Similar Reasoning Patterns

**Structured-to-structured generalization is effective.** We observe that models post-trained on math and code data exhibit strong mutual generalizability. In our observational analysis (Figure 3), models post-trained exclusively on math or code demonstrate transferable performance gains across these two domains. For example, models post-trained on math domain data achieve an average improvement of 2.18% in pass@1 on math domain tasks and 4.77% on code domain tasks. Similarly, models post-trained on code domain data improve by 9.49% in pass@1 on code domain tasks and 15.44% on math domain tasks. In both cases, the improvement is even greater on the non-finetuned domain, suggesting that math and code tasks share common structured reasoning patterns that enable RPT to generalize effectively across these domains.

**Structured reasoning patterns that are more foundational tend to exhibit stronger cross-domain transfer.** Building on the findings from our observational study, we further examine the generalizability across structured reasoning domains using interventional study results from models post-trained on single domains (Figure 4). We observe that the generalizability from math to code is notably stronger and more consistent than the reverse. This aligns with the intuition that mathematical reasoning is a more fundamental form of structured thinking, serving as the backbone for coding tasks, and thus enables better cross-domain transfer when used as RPT data. We provide a quantitative analysis of reasoning trace similarity across structured domains in Appendix B.1.

**Structured-to-unstructured generalization is limited.** Models trained on structured reasoning domains, such as math and code, exhibit substantially reduced improvements when evaluated on knowledge-intensive domain tasks. In our observational study (Figure 3), models post-trained on structured reasoning domains (i.e., math, code, or both) achieve an average improvement of -0.27% in pass@1 on knowledge-intensive domain tasks, compared to significant gains of 11.08% on math and 5.82% on code tasks. While the improvements in math and code domain tasks are statistically significant, the performance drops on knowledge-intensive reasoning tasks, indicating a lack of generalizability to unstructured domains. For instance, in the *(1)* `DeepScaleR-1.5B-Preview` and *(2)* `DeepCoder-1.5B-Preview` pair, the observed gains in math and code domain tasks are both significantly higher than those in knowledge-intensive domain tasks. Our interventional study

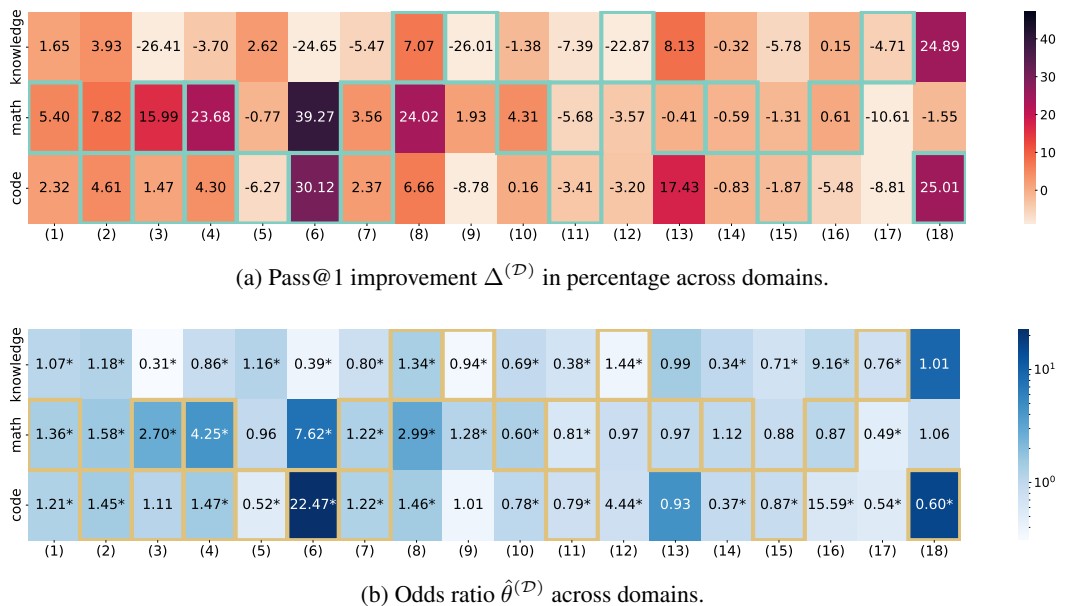

(a) Pass@1 improvement $\Delta^{(\mathcal{D})}$ in percentage across domains.

(b) Odds ratio $\hat{\theta}^{(\mathcal{D})}$ across domains.

Figure 3: Multi-domain evaluation results of existing RPT models. We highlight in-domain results with frames. RPT shows mutual generalizability between math and code, one-way transfer from knowledge-intensive reasoning to math and code, but no generalization from math or code to knowledge-intensive reasoning.

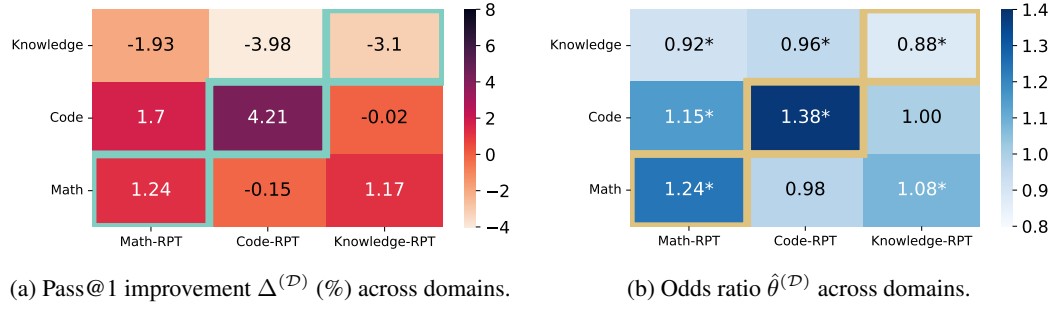

(a) Pass@1 improvement $\Delta^{(\mathcal{D})}$ (%) across domains.

(b) Odds ratio $\hat{\theta}^{(\mathcal{D})}$ across domains.

Figure 4: Multi-domain evaluation results of single domain RPT models. We highlight RPT domains with frames. RPT demonstrates generalizability from math to code and from knowledge-intensive reasoning to math, but shows no generalizability from math or code to knowledge-intensive reasoning.

results further confirm this trend (Figure 4): while the Math-RPT model shows improvements in both math and code domain tasks, its performance drops notably on knowledge-intensive domain tasks. Similarly, the Code-RPT model shows a statistically significant drop in performance on knowledge-intensive reasoning tasks. These results suggest that although structured reasoning skills generalize well across similarly structured domains, they fail to transfer effectively to domains that require less structured, more heterogeneous reasoning patterns.

**Unstructured-to-structured generalization is promising.** RPT models trained on unstructured knowledge-intensive domain data still exhibit measurable gains in structured tasks. In our observational study (Figure 3), knowledge-intensive domain RPT models show substantially higher pass@1 improvements on math (21.40%) and code (12.16%) tasks compared to tasks within the knowledge-intensive reasoning domain. Similarly, in our interventional analysis (Figure 4), the Knowledge-RPT model achieves statistically significant gains on math domain tasks and shows no noticeable degradation on code domain tasks, while underperforming on tasks within its own domain. This suggests that unstructured reasoning patterns encompass broader representational complexity and implicitly subsume the essential components of structured reasoning, functioning as a conceptual superset.

Table 3: RPT configuration variants consistently fail to improve generalizability, as shown by accuracy gains $\Delta$ (%) and odds ratios $\hat{\theta}$ on in-domain (*ID*) tasks versus out-of-domain (*OOD*) tasks across different base models and RPT algorithms.

| Base Model + RPT Algorithm | $\Delta^{(ID)} \uparrow$ | $\Delta^{(OOD)} \uparrow$ | $\Delta^{(ID)} - \Delta^{(OOD)}$ | $\hat{\theta}^{(ID)} \uparrow$ | $\hat{\theta}^{(OOD)} \uparrow$ | $\hat{\theta}^{(ID)}/\hat{\theta}^{(OOD)}$ |
|---|---|---|---|---|---|---|
| DeepSeek-R1-Distill-Qwen-1.5B + GRPO | 3.13 | -1.81 | 4.94 | 1.1881* | 0.9269* | 1.2812 |
| Llama-3.2-3B-Instruct + GRPO | 6.47 | 1.41 | 5.06 | 1.4694* | 1.0635* | 1.3817 |
| DeepSeek-R1-Distill-Qwen-1.5B + DAPO | 3.96 | -1.27 | 5.23 | 1.2395* | 0.9482* | 1.3070 |

## 4.3 Intra-domain RPT Gains Depend on Subdomain Structural Similarity

**Structured reasoning patterns generalize effectively within domain.** Consistent with prior work on math and code reasoning, our observational study shows that models post-trained on structured domains generalize well across tasks within the same domain (Figure 3). On average, models trained on math data achieve a pass@1 improvement of 2.18% on math tasks, while models trained on code data show an average improvement of 9.49% on code tasks. Our interventional analysis further confirms this trend where structured-domain models (i.e., the Code-RPT model and the Math-RPT model) exhibit the largest gains on tasks from their corresponding training domain (Figure 4). These results suggest that data following consistent and structured reasoning templates facilitates reliable generalization within the same domain, as downstream tasks can leverage similar inductive patterns.

**Unstructured reasoning patterns lack intra-domain consistency.** In contrast, models trained on knowledge-intensive domain (unstructured) data demonstrate limited or negative transfer to other unstructured tasks from different domains in our observational study (Figure 3). For instance, `Fino1-8B` (model *(12)*), post-trained on financial data, exhibits notable performance drops when evaluated on all unrelated knowledge-intensive domain tasks. Its pass@1 on `PubMedQA` (medical domain) declines from 3.26% to 1.28%, on `LegalBench` (legal domain) from 6.42% to 4.84%, and on `TabFact` declines from 64.18% to 48.39% (general tabular knowledge). Our interventional results reinforce this observation: the Knowledge-RPT model underperforms the base model on knowledge-intensive domain tasks, with the degradation in accuracy being statistically significant (Figure 4). This suggests that, unlike structured domains, unstructured reasoning tasks are highly diverse and domain-specific. They lack a shared logical template, making it difficult for RPT to generalize even within what is nominally the same domain.

We provide quantitative evidence for intra-domain diversity of different domains in Appendix B.2.

## 4.4 RPT Generalizability Remains Weak Across Configuration Variants

**RPT generalizability is consistent across different algorithms.** By comparing the in-domain and out-of-domain performance differences between GRPO and DAPO on the math domain (Table 3), we observe that both algorithms yield similar gain gaps. This suggests that despite their procedural differences, the core optimization behavior of RPT dominates, leading both algorithms to learn essentially the same domain-specific reasoning patterns.

**RPT generalizability is consistent across different base models.** By comparing the in-domain and out-of-domain performance differences between base models of DeepSeek-R1-Distill-Qwen-1.5B and Llama-3.2-3B-Instruct on the math domain (Table 3), we observe that both base models yield similar gain gaps. This suggests that the lack of generalizability in RPT is inherent to the RPT process itself, rather than a consequence of the base model architecture or pretraining data.

**RPT generalizability decreases as training steps increase.** As we show in Figure 5, the gap between in-domain and out-of-domain gains increases as the number of training steps increases and eventually stabilizes. This indicates that model generalizability gradually declines as RPT progresses. The reason is that increased exposure to the domain-specific data leads the model to overfit, improving in-domain performance while offering diminishing gains out-of-domain. The eventual convergence from a full epoch onwards occurs because the model is repeatedly trained on the same finite collection of data, limiting further changes in the gap between in-domain and out-of-domain gains. We conduct more fine-grained evaluations and analyses in Appendix H.

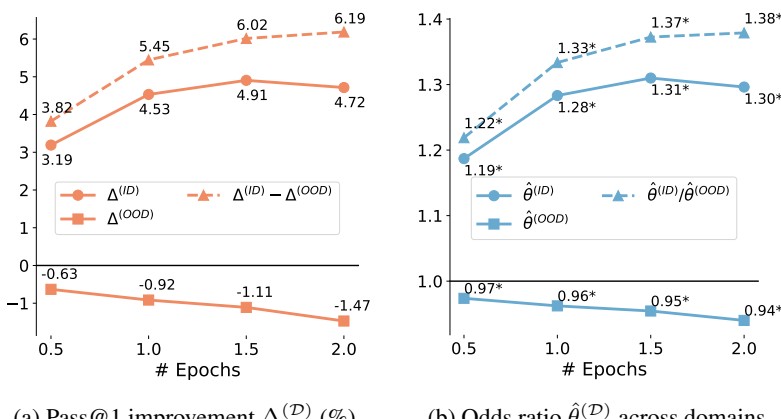

(a) Pass@1 improvement $\Delta^{(\mathcal{D})}$ (%).  (b) Odds ratio $\hat{\theta}^{(\mathcal{D})}$ across domains.

Figure 5: In-domain and out-of-domain improvements during RPT training on the math domain. The gap between in-domain and out-of-domain improvements grows as training progresses.

**Larger model sizes do not lead to better generalizability.** By comparing performance differences across model-size variants within the same model family, we find that the average in-domain gain increases by 16.5% more than the out-of-domain gain as model size grows. This pattern is consistent with the intuition that larger models are more prone to overfitting the domains on which they were RPT-trained, thereby amplifying in-domain improvements without corresponding generalization benefits. We present detailed results and analysis in Appendix C.

## 5  RELATED WORK

**RPT algorithms and frameworks.** RPT, especially RLVR, has received significant attention in building powerful reasoning models, following the release of OpenAI o1 (OpenAI et al., 2024) and DeepSeek-R1 (DeepSeek-AI et al., 2025). RPT models have proven effective for a broad range of tasks with well-defined correctness: ranging from structured tasks such as math (Xiong et al., 2025; Cui et al., 2025) and coding (Wei et al., 2025; Shojaee et al., 2023; Le et al., 2022; Shen & Zhang, 2024), to unstructured tasks like search engine use (Jin et al., 2025) and open-ended question answering (Su et al., 2025a). However, all these models are trained and evaluated on tasks within a single domain or task type. Even works on general knowledge (Su et al., 2025a) remain confined to open-ended question answering tasks, without testing transfer across fundamentally different task types. In contrast, our work directly addresses this cross-domain generalization gap.

**Limitations of RPT.** Despite the recent successes of RPT in improving language model reasoning capabilities, the limitations of RPT in general have been widely studied. At the training phase, SFT with LLM reasoning traces, without RL, has been shown to be effective enough (Team, 2025b; Yue et al., 2025). At the inference phase, the quality of reasoning models depends crucially on their ability to scale under test-time compute constraints (Muennighoff et al., 2025; Zhao et al., 2025b). Moreover, the effectiveness of LLMs' lengthy "thinking" processes has also been challenged (Ma et al., 2025; Ye et al., 2025). Building on these observations, our work aims to examine the limitations of RPT at a finer granularity, specifically its data generalizability across the training and inference phases.

## 6  CONCLUSION

In this paper, we identify important limitations in the generalizability of RPT across domains. Through both observational and interventional studies, we consistently find that while RPT produces substantial improvements within training domains, its generalization to unseen domains is limited. In particular, while there is evidence of cross-domain transfer between structured domains like math and code, there is little evidence of transfer to unstructured domains. Our work emphasizes the need for a more nuanced understanding of cross-domain knowledge transfer in LLMs.

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

## A  USE OF LARGE LANGUAGE MODELS (LLMS)

No LLMs were used in the ideation, writing, or preparation of this paper. All content was conceived, drafted, and revised solely by the authors.

## B  REASONING TEMPLATE ANALYSIS

To better interpret the generalizability across domains, we analyze the reasoning templates used in different types of tasks. Specifically, we randomly sampled $\min(\text{dataset size}, 100)$ tasks from each evaluation benchmark, resulting in 1,470 task instances in total. We then used Claude Sonnet 4.5 (Anthropic, 2025), the state-of-the-art periphery reasoning model with exposed reasoning traces, to complete the selected tasks. Next, we collected the generated reasoning traces and used GPT-4o (OpenAI, 2024) to tag each step of the reasoning traces using the following taxonomy:

- **READ_RESTATE**: Restating or understanding the question.
- **SETUP**: Defining variables, listing assumptions, establishing context, or initializing structures.
- **PLAN**: High-level strategy, decomposition, or outlining an algorithm.
- **EXECUTE_STEP**: Any intermediate logical, mathematical, algorithmic, computational, or transformational operation.
- **CONTROL_FLOW**: Branching, case analysis, loops, recursion, or considering alternative scenarios.
- **VERIFY**: Testing, checking correctness, or performing sanity checks.
- **FINAL_ANSWER**: Final conclusion or solution.
- **OTHER**: Anything not fitting the categories above.

Finally, we quantitatively analyzed the divergence of the reasoning templates in the tagged traces within and across domains. We used the Jeffreys divergence (Jeffreys, 1961) to measure the symmetric divergence between step-type distributions.

We present the detailed distribution of each domain in Figure 6. We observe that the annotations are of high quality: almost no steps are labeled as OTHER, indicating that our tagset provides comprehensive coverage of the reasoning process. Building on this, we find that EXECUTE_STEP is the most dominant category across all three domains. PLAN is particularly important in math and code tasks, reflecting their structured, multi-step problem-solving nature, while SETUP plays a comparatively larger role in knowledge-intensive tasks, where establishing context or recalling background information is crucial.

We now use the distribution vectors to quantify our hypothesis in Section 4.

### B.1  MATH AND CODE DOMAINS SHARE SIMILAR REASONING TEMPLATES (§4.2)

We computed the Jeffreys divergence (Jeffreys, 1961) between two domain-level reasoning–step distributions, which is obtained by aggregating the tagged traces of each domain and normalizing single-domain traces into probability distributions. Using these aggregated distributions, the Jeffreys divergences are 0.18 (between math and code), 0.29 (between math and knowledge-intensive), and 0.69 (between code and knowledge-intensive). This indicates that math and code share more similar reasoning templates, while knowledge-intensive tasks require substantially different reasoning.

### B.2  KNOWLEDGE-INTENSIVE DOMAIN HAS LARGER INTRA-DOMAIN VARIANCE (§4.3)

We compute the Jeffreys divergence between every pair of datasets within each domain, using the normalized reasoning–step distributions for each dataset. Averaging over all pairwise comparisons, the Jeffreys divergences 0.15 (within the math domain), 0.14 (within the code domain), and 0.19 (within the knowledge-intensive domain). This indicates that knowledge-intensive tasks exhibit a more diverse set of reasoning templates compared to math and code tasks.

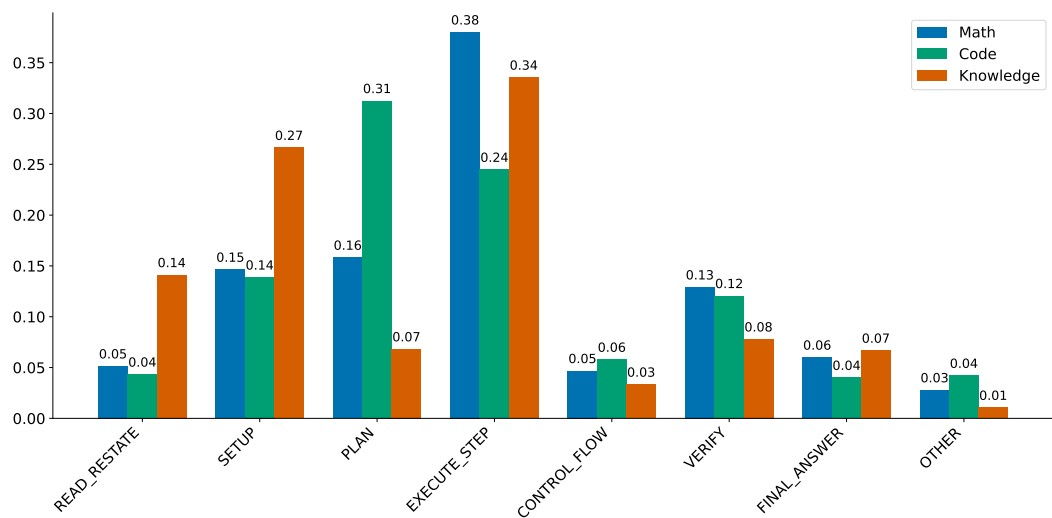

Figure 6: Distribution of reasoning-step tags across domains.

Table 4: Larger model size does not improve generalizability, as shown by accuracy gains $\Delta$ (%) and odds ratios $\hat{\theta}$ on in-domain (*ID*) tasks compared to out-of-domain (*OOD*) tasks across different size variants. For difference-based metrics, the model size effect is computed as the average difference between the larger and smaller variant (Large – Small); for ratio-based metrics, the model size effect is computed as the average ratio (Large / Small).

| **Model** | $\Delta^{(ID)} \uparrow$ | $\Delta^{(OOD)} \uparrow$ | $\Delta^{(ID)} - \Delta^{(OOD)}$ | $\hat{\theta}^{(ID)} \uparrow$ | $\hat{\theta}^{(OOD)} \uparrow$ | $\hat{\theta}^{(ID)}/\hat{\theta}^{(OOD)}$ |
|---|---|---|---|---|---|---|
| DeepCoder-1.5B | 4.61 | 4.01 | 0.60 | 1.4496* | 1.1840* | 1.2243 |
| DeepCoder-14B | -1.87 | -5.68 | 3.81 | 0.8715* | 0.7122* | 1.2238 |
| OREAL-7B | -0.41 | 8.44 | -8.85 | 0.9714 | 1.4724* | 0.6597 |
| OREAL-32B | 0.61 | -0.04 | 0.65 | 1.0571 | 0.9982 | 1.0590 |
| Fino1-8B | -3.84 | -27.89 | 24.05 | 0.8319* | 0.3034* | 2.7420 |
| Fin-o1-14B | -3.60 | -5.43 | 1.83 | 0.8481* | 0.7047* | 1.2034 |
| AZR-Coder-3B | -6.27 | 2.55 | -8.82 | 0.5235* | 1.1517* | 2.7420 |
| AZR-Coder-7B | 30.12 | -38.80 | 68.92 | 22.4671* | 0.2452* | 91.6456 |
| AZR-Coder-14B | 25.01 | 24.34 | 0.67 | 15.5878* | 8.5023* | 8.5023 |
| **Model Size Effects** | **9.56** | **7.98** | **1.58** | **12.683** | **7.645** | **6.610** |

## C   RPT GENERALIZABILITY ON MODELS ACROSS DIFFERENT SIZES

By examining the performance differences across model-size variants in Table 4, we observe no clear trend indicating improved generalizability with larger models. Several models exhibit reduced generalizability as size increases. We believe this is because larger models more effectively overfit to the domains on which they were RPT-trained. The primary exception is the Fino family, whose improved generalizability is explained by a change in base model rather than model size itself (Fino1-8B is based on Llama, whereas Fin-o1-14B uses Qwen3-14B).

## D   REWARD SIGNALS DURING INTERVENTIONAL STUDY

### D.1   REWARD DEFINITIONS

Following prior work that has demonstrated promising performance (Luo et al., 2025a;b; Yu et al., 2025), we use domain-specific binary reward functions:

- **Math:** 1 if model's answer is mathematically equivalent to the ground truth; otherwise 0.

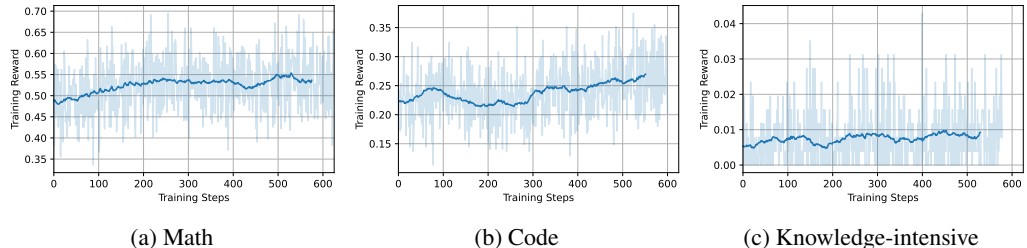

(a) Math             (b) Code           (c) Knowledge-intensive

Figure 7: Training rewards of our RPT training runs for math, code, and knowledge-intensive domains. Over the 625-step training process, the 50-step moving average training rewards of math, code, and knowledge-intensive domains increase by 9.7%, 21.7%, and 71.0%, respectively.

Table 5: Mixed-domain RPT performance compared to the best single-domain RPT model. The effects column reports differences for $\Delta^{(D)}$ (Mixed $-$ Min) and ratios for $\hat{\theta}^{(D)}$ (Mixed / Min).

|  | Mixed-RPT | Min(RPT) | Effects |
|---|---|---|---|
| $\Delta^{(\text{Knowledge})}$ ↑ | -2.17 | -3.98 | 1.81 |
| $\Delta^{(\text{Code})}$ ↑ | 1.91 | -0.02 | 1.93 |
| $\Delta^{(\text{Math})}$ ↑ | -3.62 | -0.15 | -3.47 |
| $\hat{\theta}^{(\text{Knowledge})}$ ↑ | 0.91* | 0.88* | 1.03 |
| $\hat{\theta}^{(\text{Code})}$ ↑ | 1.18* | 1.00 | 1.18 |
| $\hat{\theta}^{(\text{Math})}$ ↑ | 0.83* | 0.98 | 0.85 |

- **Code:** 1 if model's answer passes all unit tests; otherwise 0.
- **Knowledge:** 1 if model's answer matches the ground truth string; otherwise 0.

For extraction, we use symbolic and string-based equivalence checking for math via the `math-verify` package (Kydlíček & Gandenberger, 2025), unit tests for code, and string-based checking for general knowledge. To quantitatively assess reward quality, we sampled 100 training examples and manually verified the correctness of their reward signals. The correctness rates for math, code, and general knowledge are 94%, 99%, and 97%, respectively, indicating consistently high reward quality across domains.

### D.2 OPTIMIZATION DYNAMICS

In this section, we show the optimization dynamics of our RPT training runs. We present the training rewards for each domain in Figure 7. For each domain, we show the raw training rewards and the moving average rewards over 50-step windows.

As shown in the figures, we find that the training rewards of all domains exhibit noticeable increase over the course of RPT training. Quantitative, the 50-step average training rewards of math, code, and knowledge-intensive domains increase by 9.7%, 21.7%, and 71.0%. This shows that the model learned domain-specific reasoning techniques in single-domain RPTs, which in turn improves the task-specific performance.

### E MIXED DOMAIN TRAINING

We train an additional model using a mixed-domain dataset, created by randomly sampling 13,333 data points from each of the single-domain datasets. As shown in Table 5, the mixed-domain RPT model achieves an overall 2.7 percentage-point accuracy improvement compared to individual single-domain RPT models.Importantly, this mixed-domain result should be interpreted as a lower bound: because we simply drew a small, uniformly sampled subset from each domain without any optimization, a more principled mixed-domain curation strategy could potentially yield even stronger performance.

# F PER MODEL PER TASK RESULTS

## F.1 OBSERVATIONAL STUDY RESULTS

Table 6: Observational study results. The model IDs corresponds to Table 1.

| Model | pubmedqa | medqa | aime2024 | gsm8k | math500 | amc23 | tab_fact | legalbench | finben | livecodebench | codeforces | polyglot | humaneval | bigcodebench | mbpp | usaco |
|---|---|---|---|---|---|---|---|---|---|---|---|---|---|---|---|---|
| 1-RPT | 0.640 | 0.242 | 0.331 | 0.823 | 0.892 | 0.697 | 0.694 | 0.597 | 0.547 | 0.230 | 0.033 | 0.009 | 0.713 | 0.126 | 0.511 | 0.065 |
| 1-Base | 0.616 | 0.261 | 0.279 | 0.747 | 0.842 | 0.683 | 0.668 | 0.586 | 0.520 | 0.175 | 0.024 | 0.000 | 0.652 | 0.112 | 0.474 | 0.052 |
| 2-RPT | 0.632 | 0.242 | 0.375 | 0.848 | 0.906 | 0.711 | 0.714 | 0.617 | 0.580 | 0.235 | 0.043 | 0.013 | 0.701 | 0.130 | 0.584 | 0.055 |
| 2-Base | 0.616 | 0.261 | 0.279 | 0.747 | 0.842 | 0.683 | 0.668 | 0.586 | 0.520 | 0.175 | 0.024 | 0.000 | 0.652 | 0.112 | 0.474 | 0.052 |
| 3-RPT | 0.214 | 0.025 | 0.096 | 0.888 | 0.798 | 0.497 | 0.071 | 0.339 | 0.176 | 0.150 | 0.027 | 0.049 | 0.780 | 0.425 | 0.647 | 0.029 |
| 3-Base | 0.326 | 0.266 | 0.029 | 0.766 | 0.464 | 0.323 | 0.642 | 0.611 | 0.316 | 0.110 | 0.009 | 0.009 | 0.640 | 0.397 | 0.675 | 0.016 |
| 4-RPT | 0.710 | 0.355 | 0.177 | 0.917 | 0.774 | 0.648 | 0.688 | 0.527 | 0.524 | 0.188 | 0.021 | 0.009 | 0.543 | 0.172 | 0.621 | 0.033 |
| 4-Base | 0.714 | 0.401 | 0.062 | 0.812 | 0.080 | 0.405 | 0.650 | 0.550 | 0.620 | 0.110 | 0.004 | 0.000 | 0.543 | 0.079 | 0.589 | 0.013 |
| 5-RPT | 0.508 | 0.250 | 0.010 | 0.424 | 0.370 | 0.125 | 0.539 | 0.200 | 0.360 | 0.052 | 0.003 | 0.009 | 0.030 | 0.073 | 0.560 | 0.000 |
| 5-Base | 0.474 | 0.211 | 0.006 | 0.434 | 0.412 | 0.110 | 0.460 | 0.159 | 0.388 | 0.043 | 0.003 | 0.013 | 0.268 | 0.179 | 0.673 | 0.003 |
| 6-RPT | 0.592 | 0.395 | 0.046 | 0.801 | 0.662 | 0.369 | 0.761 | 0.175 | 0.504 | 0.128 | 0.014 | 0.013 | 0.378 | 0.382 | 0.768 | 0.029 |
| 6-Base | 0.622 | 0.233 | 0.008 | 0.114 | 0.496 | 0.141 | 0.228 | 0.617 | 0.580 | 0.013 | 0.001 | 0.004 | 0.104 | 0.068 | 0.005 | 0.007 |
| 7-RPT | 0.668 | 0.265 | 0.317 | 0.804 | 0.864 | 0.683 | 0.709 | 0.600 | 0.266 | 0.200 | 0.036 | 0.004 | 0.640 | 0.102 | 0.552 | 0.055 |
| 7-Base | 0.616 | 0.261 | 0.279 | 0.747 | 0.842 | 0.683 | 0.668 | 0.586 | 0.520 | 0.175 | 0.024 | 0.000 | 0.652 | 0.112 | 0.474 | 0.052 |
| 8-RPT | 0.320 | 0.316 | 0.562 | 0.718 | 0.952 | 0.745 | 0.460 | 0.628 | 0.603 | 0.385 | 0.170 | 0.000 | 0.848 | 0.357 | 0.597 | 0.388 |
| 8-Base | 0.326 | 0.266 | 0.029 | 0.766 | 0.464 | 0.323 | 0.642 | 0.611 | 0.316 | 0.110 | 0.009 | 0.009 | 0.640 | 0.397 | 0.675 | 0.016 |
| 9-RPT | 0.724 | 0.501 | 0.015 | 0.636 | 0.448 | 0.151 | 0.615 | 0.220 | 0.587 | 0.050 | 0.004 | 0.004 | 0.189 | 0.109 | 0.129 | 0.007 |
| 9-Base | 0.732 | 0.496 | 0.008 | 0.695 | 0.224 | 0.120 | 0.616 | 0.603 | 0.662 | 0.060 | 0.008 | 0.000 | 0.573 | 0.261 | 0.271 | 0.000 |
| 10-RPT | 0.620 | 0.245 | 0.294 | 0.809 | 0.862 | 0.727 | 0.650 | 0.558 | 0.541 | 0.175 | 0.021 | 0.000 | 0.622 | 0.082 | 0.518 | 0.039 |
| 10-Base | 0.616 | 0.261 | 0.279 | 0.747 | 0.842 | 0.683 | 0.668 | 0.586 | 0.520 | 0.175 | 0.024 | 0.000 | 0.652 | 0.112 | 0.474 | 0.052 |
| 11-RPT | 0.740 | 0.304 | 0.367 | 0.911 | 0.924 | 0.825 | 0.791 | 0.690 | 0.603 | 0.311 | 0.063 | 0.004 | 0.817 | 0.251 | 0.810 | 0.143 |
| 11-Base | 0.740 | 0.324 | 0.581 | 0.914 | 0.940 | 0.908 | 0.808 | 0.748 | 0.738 | 0.328 | 0.094 | 0.027 | 0.817 | 0.320 | 0.826 | 0.186 |
| 12-RPT | 0.128 | 0.304 | 0.029 | 0.753 | 0.414 | 0.227 | 0.484 | 0.287 | 0.277 | 0.018 | 0.002 | 0.049 | 0.543 | 0.360 | 0.630 | 0.010 |
| 12-Base | 0.326 | 0.266 | 0.029 | 0.766 | 0.464 | 0.323 | 0.642 | 0.611 | 0.316 | 0.110 | 0.009 | 0.009 | 0.640 | 0.397 | 0.675 | 0.016 |
| 13-RPT | 0.368 | 0.110 | 0.263 | 0.898 | 0.802 | 0.728 | 0.276 | 0.566 | 0.361 | 0.068 | 0.010 | 0.018 | 0.494 | 0.164 | 0.809 | 0.016 |
| 13-Base | 0.144 | 0.098 | 0.273 | 0.911 | 0.816 | 0.700 | 0.086 | 0.561 | 0.138 | 0.050 | 0.006 | 0.018 | 0.354 | 0.138 | 0.204 | 0.013 |
| 14-RPT | 0.614 | 0.231 | 0.263 | 0.754 | 0.846 | 0.650 | 0.626 | 0.584 | 0.529 | 0.147 | 0.025 | 0.009 | 0.622 | 0.099 | 0.468 | 0.036 |
| 14-Base | 0.616 | 0.261 | 0.279 | 0.747 | 0.842 | 0.683 | 0.668 | 0.586 | 0.520 | 0.175 | 0.024 | 0.000 | 0.652 | 0.112 | 0.474 | 0.052 |
| 15-RPT | 0.779 | 0.791 | 0.606 | 0.947 | 0.794 | 0.927 | 0.887 | 0.821 | 0.500 | 0.357 | 0.110 | 0.000 | 0.896 | 0.259 | 0.889 | 0.238 |
| 15-Base | 0.775 | 0.781 | 0.652 | 0.951 | 0.784 | 0.952 | 0.893 | 0.819 | 0.720 | 0.400 | 0.137 | 0.000 | 0.872 | 0.259 | 0.896 | 0.345 |
| 16-RPT | 0.552 | 0.183 | 0.444 | 0.941 | 0.898 | 0.839 | 0.224 | 0.690 | 0.337 | 0.165 | 0.039 | 0.000 | 0.104 | 0.000 | 0.724 | 0.085 |
| 16-Base | 0.585 | 0.156 | 0.421 | 0.948 | 0.886 | 0.823 | 0.230 | 0.689 | 0.332 | 0.177 | 0.039 | 0.000 | 0.652 | 0.154 | 0.718 | 0.094 |
| 17-RPT | 0.720 | 0.683 | 0.221 | 0.923 | 0.750 | 0.664 | 0.813 | 0.776 | 0.659 | 0.282 | 0.065 | 0.004 | 0.598 | 0.246 | 0.807 | 0.147 |
| 17-Base | 0.763 | 0.825 | 0.675 | 0.951 | 0.504 | 0.945 | 0.904 | 0.820 | 0.695 | 0.465 | 0.156 | 0.000 | 0.945 | 0.230 | 0.893 | 0.423 |
| 18-RPT | 0.632 | 0.401 | 0.023 | 0.083 | 0.180 | 0.228 | 0.647 | 0.107 | 0.606 | 0.113 | 0.007 | 0.009 | 0.207 | 0.184 | 0.817 | 0.036 |
| 18-Base | 0.076 | 0.052 | 0.002 | 0.245 | 0.056 | 0.078 | 0.043 | 0.040 | 0.060 | 0.000 | 0.000 | 0.018 | 0.152 | 0.050 | 0.050 | 0.003 |

## F.2 INTERVENTIONAL STUDY RESULTS

Table 7: Full interventional evaluation results across models and benchmarks.

| Model | pubmedqa | medqa | aime2024 | gsm8k | math500 | amc23 | tab_fact | legalbench | finben | livecodebench | codeforces | polyglot | humaneval | bigcodebench | mbpp | usaco |
|---|---|---|---|---|---|---|---|---|---|---|---|---|---|---|---|---|
| Math-RPT | 0.636 | 0.233 | 0.319 | 0.789 | 0.871 | 0.688 | 0.678 | 0.585 | 0.447 | 0.203 | 0.034 | 0.013 | 0.689 | 0.109 | 0.514 | 0.055 |
| Code-RPT | 0.662 | 0.222 | 0.296 | 0.700 | 0.847 | 0.666 | 0.664 | 0.591 | 0.472 | 0.158 | 0.024 | 0.009 | 0.530 | 0.253 | 0.540 | 0.000 |
| Knowledge-RPT | 0.636 | 0.221 | 0.323 | 0.757 | 0.852 | 0.688 | 0.658 | 0.584 | 0.414 | 0.183 | 0.023 | 0.009 | 0.591 | 0.100 | 0.493 | 0.036 |
| Math-RPT-DAPO | 0.647 | 0.282 | 0.333 | 0.818 | 0.724 | 0.770 | 0.664 | 0.585 | 0.475 | 0.172 | 0.030 | 0.004 | 0.585 | 0.070 | 0.496 | 0.042 |
| Math-RPT-Llama | 0.518 | 0.535 | 0.087 | 0.824 | 0.518 | 0.372 | 0.662 | 0.585 | 0.300 | 0.085 | 0.008 | 0.000 | 0.530 | 0.143 | 0.395 | 0.003 |
| Math-RPT-0.5-epoch | 0.660 | 0.306 | 0.333 | 0.804 | 0.730 | 0.759 | 0.661 | 0.585 | 0.500 | 0.200 | 0.028 | 0.000 | 0.585 | 0.066 | 0.485 | 0.039 |
| Math-RPT-1-epoch | 0.632 | 0.308 | 0.321 | 0.832 | 0.726 | 0.775 | 0.662 | 0.584 | 0.491 | 0.190 | 0.026 | 0.004 | 0.610 | 0.063 | 0.496 | 0.042 |
| Math-RPT-1.5-epochs | 0.655 | 0.315 | 0.340 | 0.847 | 0.726 | 0.747 | 0.662 | 0.586 | 0.477 | 0.188 | 0.024 | 0.004 | 0.659 | 0.073 | 0.516 | 0.046 |
| Math-RPT-2-epochs | 0.646 | 0.284 | 0.331 | 0.838 | 0.728 | 0.762 | 0.658 | 0.585 | 0.469 | 0.200 | 0.028 | 0.000 | 0.604 | 0.069 | 0.487 | 0.052 |
| Mixed-RPT | 0.666 | 0.285 | 0.227 | 0.727 | 0.660 | 0.739 | 0.679 | 0.580 | 0.448 | 0.207 | 0.040 | 0.004 | 0.634 | 0.064 | 0.564 | 0.052 |
| Base: Qwen | 0.616 | 0.261 | 0.279 | 0.747 | 0.842 | 0.683 | 0.668 | 0.586 | 0.520 | 0.175 | 0.024 | 0.000 | 0.652 | 0.112 | 0.474 | 0.052 |
| Base: Llama | 0.467 | 0.201 | 0.044 | 0.788 | 0.444 | 0.239 | 0.553 | 0.587 | 0.285 | 0.080 | 0.005 | 0.000 | 0.494 | 0.153 | 0.561 | 0.007 |

# G    RELATIVE IMPROVEMENT

To address the importance of relative performance gains, we complement our absolute aggregated improvement measure with a secondary metric that captures proportional improvements while remaining well-defined even when the base model attains zero accuracy.

Formally, we report the *relative aggregated improvement*:

$$\widetilde{\Delta}_{i,j}^{(\mathcal{D})} = \frac{\sum_{t\in\mathcal{D}} N_t R_t \cdot \rho_{i,j,t}}{\sum_{t\in\mathcal{D}} N_t R_t}, \tag{1}$$

where the per-benchmark relative gain is defined as

$$\rho_{i,j,t} = \begin{cases} \dfrac{A_{i,t} - A_{j,t}}{A_{j,t}}, & A_{j,t} > 0, \\ A_{i,t} - A_{j,t}, & A_{j,t} = 0. \end{cases} \tag{2}$$

We now include all the results we reported with relative improvements. We can see that our conclusion is further strengthened with this secondary metric. We illustrate our findings as follows.

**The gap between in-domain and out-of-domain tasks is more evident under relative improvements.** Table 9 compares the in-domain and out-of-domain Pass@1 relative improvements of model pairs in the observational study. The relative gap is 256.71%, indicating that accuracy on in-domain tasks increases more than $3.5\times$ on out-of-domain tasks. This contrast is substantially sharper than what is shown by the primary metrics in Table 2, where the absolute accuracy improvement is 6.07%.

The same pattern appears consistently across all settings. Relative improvements highlight larger separations between domains in the observational study (Figure 8) compared to the accuracy-based results (Figure 3). The effect also holds for the interventional study when comparing Figures 9 and 10 with their absolute counterparts in Figures 2 and 4.

Overall, relative improvement serves as a complementary metric that amplifies domain-wise gaps of RPT model performance, and these clearer separations further strengthen our conclusion that RPT lacks generalizability across domains.

Table 8: Relative Pass@1 improvements (%) for in-domain and out-of-domain tasks across model pairs in the observational study.

| Metric | (1) | (2) | (3) | (4) | (5) | (6) | (7) | (8) | (9) | (10) | (11) | (12) | (13) | (14) | (15) | (16) | (17) | (18) | Avg. |
|---|---|---|---|---|---|---|---|---|---|---|---|---|---|---|---|---|---|---|---|
| $\widetilde{\Delta}^{(ID)}$ | 9.04 | 34.78 | 68.34 | 665.49 | -27.39 | 5030.48 | 6.08 | 23.51 | -42.86 | 6.37 | -8.42 | -38.91 | -0.72 | -1.50 | -8.34 | 1.21 | -5.86 | 540.42 | **347.32** |
| $\widetilde{\Delta}^{(OOD)}$ | 3.42 | 7.06 | -44.57 | 672.06 | 16.04 | -17.46 | -9.92 | 526.15 | -16.14 | -2.49 | -10.32 | -11.83 | 66.36 | -0.65 | -7.90 | -0.55 | -19.77 | 481.47 | **90.61** |
| $\widetilde{\Delta}^{(ID)} - \widetilde{\Delta}^{(OOD)}$ | 5.62 | 27.72 | 112.91 | -6.58 | -43.43 | 5047.94 | 16.00 | -502.63 | -26.72 | 8.86 | 1.90 | -27.08 | -67.07 | -0.85 | -0.44 | 1.77 | 13.91 | 58.94 | **256.71** |

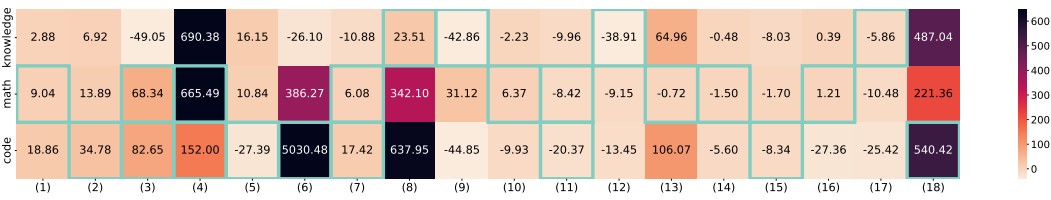

Figure 8: Relative Pass@1 improvements (%) across domains for model pairs in the observational study.

**The major RPT configuration for interventional analysis achieves the strongest generalizability.** As shown in Table 9, our main configuration (Qwen + GRPO) exhibits a substantially smaller relative accuracy gap than the other two variants. This contrast becomes especially clear under the relative-improvement metric: the Llama-based configuration yields a gap of 28.38%, whereas the Qwen-based configuration yields only 8.75%. By comparison, the absolute accuracy gaps reported in Table 3 differ only slightly (4.94% vs. 5.23%). These results demonstrate that adjusting RPT implementation variants does not improve generalizability. They also validate the design and experimental setup of our interventional study.

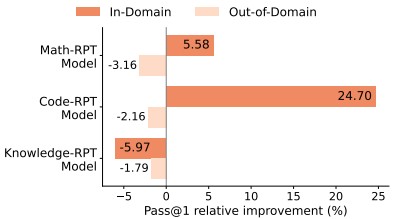

Figure 9: Relative Pass@1 improvements (%) for in-domain and out-of-domain tasks across model pairs in the interventional study.

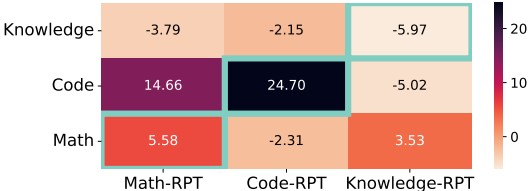

Figure 10: Relative Pass@1 improvements (%) across domains for model pairs in the interventional study.

Table 9: RPT configuration variants consistently fail to improve generalizability, as shown by relative Pass@1 improvements $\widetilde{\Delta}$ (%) on in-domain (*ID*) tasks versus out-of-domain (*OOD*) tasks across different base models and RPT algorithms.

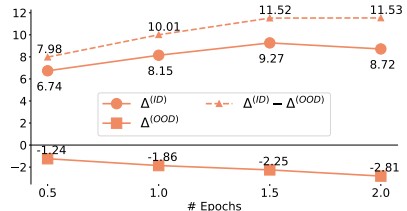

Figure 11: Relative Pass@1 improvements (%) across checkpoints.

| Base Model + RPT Algorithm | $\widetilde{\Delta}^{(ID)}$ | $\widetilde{\Delta}^{(OOD)}$ | $\widetilde{\Delta}^{(ID)} - \widetilde{\Delta}^{(OOD)}$ |
|---|---|---|---|
| DeepSeek-R1-Distill-Qwen-1.5B + GRPO | 5.58 | -3.16 | 8.75 |
| Llama-3.2-3B-Instruct + GRPO | 32.96 | 4.58 | 28.38 |
| DeepSeek-R1-Distill-Qwen-1.5B + DAPO | 7.81 | -2.43 | 10.24 |

**The effect of training epochs becomes more evident, and the convergence trend is clearer with relative improvements.** As shown in Figure 11, the relative performance first increases and then stabilizes as the number of RPT epochs grows. Compared to the two primary metrics reported in Figure 5, the relative-improvement view makes both the growth phase and the convergence pattern more pronounced. This further highlights how the model increasingly overfits to the training domain and lacks generalizability.

# H CHECKPOINT EVALUATIONS AT 100-STEP INTERVALS

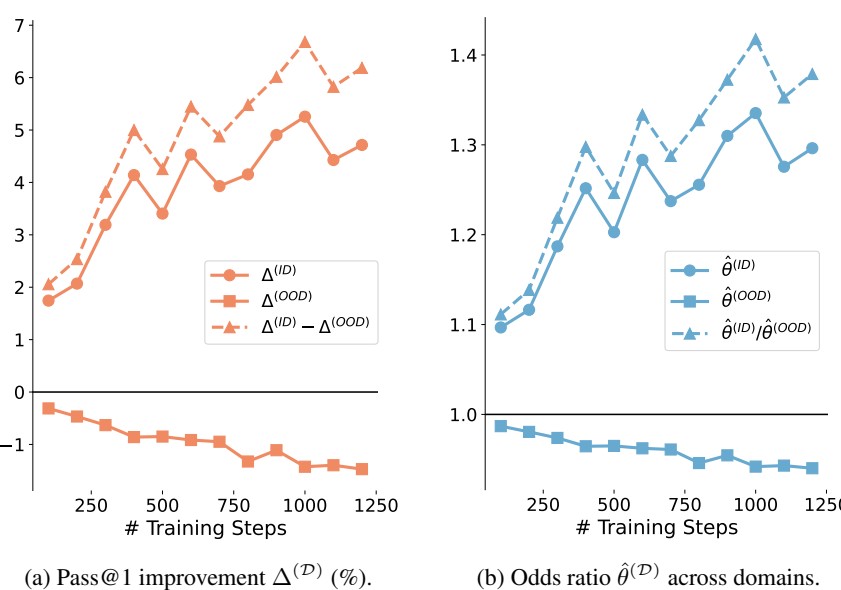

(a) Pass@1 improvement $\Delta^{(\mathcal{D})}$ (%).  (b) Odds ratio $\hat{\theta}^{(\mathcal{D})}$ across domains.

Figure 12: In-domain and out-of-domain improvements at 100-step intervals during RPT training on the math domain. A full epoch corresponds to 600 training steps. The gap between in-domain and out-of-domain improvements grows as training progresses. We report the specific values in Table 10.

We perform a finer-grained analysis of model behavior by evaluating intermediate checkpoints at 100-step intervals throughout RPT training. We display the results in Figure 12 and Table 10. These results further strengthen our conclusion in Section 4.4 that as training progresses, the generalizability of RPT decreases. Specifically, we observe that:

1. in-domain gains remain consistently positive, and out-of-domain gains remain consistently negative;

2. in-domain gains increase, whereas out-of-domain gains decrease;

3. the gap between in-domain and out-of-domain gains widens as training advances; and

4. the trends eventually stabilize, particularly after the end of the first epoch, when the model has been exposed to the full training dataset.

Table 10: Math-RPT checkpoint performance at 100-step intervals, reporting accuracy gains $\Delta$ (%) and odds ratios $\hat{\theta}$ for in-domain (*ID*) and out-of-domain (*OOD*) tasks. A full epoch corresponds to 600 training steps.

| Model | $\Delta^{(ID)} \uparrow$ | $\Delta^{(OOD)} \uparrow$ | $\Delta^{(ID)} - \Delta^{(OOD)}$ | $\hat{\theta}^{(ID)} \uparrow$ | $\hat{\theta}^{(OOD)} \uparrow$ | $\hat{\theta}^{(ID)}/\hat{\theta}^{(OOD)}$ |
|---|---|---|---|---|---|---|
| Math-RPT-100-steps | 1.74 | -0.31 | 2.06 | 1.0968 | 0.9869 | 1.1113 |
| Math-RPT-200-steps | 2.07 | -0.47 | 2.54 | 1.1164 | 0.9806* | 1.1386 |
| Math-RPT-300-steps | 3.19 | -0.63 | 3.82 | 1.1870* | 0.9738* | 1.2189 |
| Math-RPT-400-steps | 4.14 | -0.86 | 5.00 | 1.2517* | 0.9645* | 1.2977 |
| Math-RPT-500-steps | 3.41 | -0.85 | 4.26 | 1.2027* | 0.9649* | 1.2464 |
| Math-RPT-600-steps | 4.53 | -0.92 | 5.45 | 1.2833* | 0.9623* | 1.3336 |
| Math-RPT-700-steps | 3.93 | -0.95 | 4.88 | 1.2373* | 0.9609* | 1.2876 |
| Math-RPT-800-steps | 4.15 | -1.33 | 5.48 | 1.2556* | 0.9459* | 1.3275 |
| Math-RPT-900-steps | 4.91 | -1.11 | 6.02 | 1.3100* | 0.9544* | 1.3725 |
| Math-RPT-1000-steps | 5.26 | -1.43 | 6.68 | 1.3353* | 0.9419* | 1.4177 |
| Math-RPT-1100-steps | 4.43 | -1.40 | 5.82 | 1.2756* | 0.9430* | 1.3527 |
| Math-RPT-1200-steps | 4.72 | -1.47 | 6.19 | 1.2962* | 0.9401* | 1.3789 |

