# OpenReview forum: "Breaking Barriers: Do Reinforcement Post Training Gains Transfer To Unseen Domains?"
_ICLR.cc/2026/Conference — ICLR 2026 Poster_

### Official Review · Reviewer_FGBZ · 2025-10-28

**Soundness:** 3
**Presentation:** 3
**Contribution:** 3
**Rating:** 6
**Confidence:** 4

**Summary:**

This paper looks at how reinforcement learning post-training generalizes to out-of-domain evaluations. The authors curate various models on huggingface trained with RL, and train their own models on specific domains. They find that RL post-training does not seem to generalize to arbitrary unseen domains, but does show generalization between math and code domains, which the authors hypothesise is due to similar reasoning templates being applicable in the two domains.

**Strengths:**

- A wide range of models and evaluations are studied, making the findings seem fairly robust (especially with the intervention experiments).
- The evaluation itself is well-performed, using appropriate metrics and statistical testing.
- Paper is clear and reasonably structured, and the findings appear useful (especially that knowledge-focused tasks do not seem to transfer well to/from math and code settings).

**Weaknesses:**

- It appears that the open source models do not uniformly do better IID than OOD, for example, model 4 (Eurus-Prime) does 15 points better OOD than IID! Do you have explanations for this beyond ‘differences in implementation details’? It would be useful to have some idea of why there is this variance between models - the ID-OOD gap has a standard deviation of ~18 in table 2! I believe the trends described hold, but it would be good to have some idea of why there is this variance across models.
- It seems that knowledge-RPT drops performance on the knowledge tasks - could this be more due to a domain mismatch between the knowledge-RPT data and evaluation tasks? The evaluations are very domain specific (medical QA, legal QA), while the training data is from multi-subject RLVR, which covers a much broader set of domains.
- It would be useful to quantify/test the hypothesis in section 4.3 more thoroughly: if the reasoning templates are similar between code and math, could you examine some samples or measure overlap between reasoning chains to test this hypothesis? Looking only at downstream numbers does not fully explain what is happening. For example, it may be that only smaller subsets of the code data are similar to the math data, or that there is some cross-domain contamination between the two sets (e.g., code questions that require doing math, or math problems that require writing code).
- For the intervention experiments, I’d be curious to see if the base model is a potential confounder. The deepseek distil model used has been extensively trained on math data, so it may be that this makes it less easy to adapt to knowledge tasks, or better primed to improve math performance when trained on code data.

Overall, I think this is a solid paper, although its scope is somewhat limited. It would be useful to get more justifications around the knowledge-RPT setting and some discussion around the variance in the observational results.

**Questions:**

See weaknesses.

---

> ### Author Response · Authors · 2025-11-21
> **Response to Reviewer FGBZ (1/2): W1**
>
> Dear Reviewer FGBZ,
>
> Thank you for your detailed summary and constructive feedback. We are glad that you find our work robust, well-formed, and useful. We address your comments as follows.
>
> **W1 (variance in IID–OOD gaps across models):** First, we sincerely appreciate your pointing out the outlier behavior of Eurus, which prompted us to re-examine our evaluation setup in detail. Upon revisiting the model configuration, we discovered an important oversight: although the Hugging Face documentation for PRIME-RL/Eurus-2-7B-PRIME reports the base model as Qwen2.5-Math-7B, there is an additional intermediate SFT stage trained on science data before RPT. This SFT step substantially boosts performance on OOD domains, which explains the unusually large OOD gains observed in the original submission. **To correct this, we have updated our experiments to use the actual model that serves as the starting point for RPT (PRIME-RL/Eurus-2-7B-SFT) as the proper “base model” for comparison.** After rerunning all Eurus evaluations with the correct initialization and conducting a full pass on all model pairs to verify consistency, we updated Table 2, Figure 3, and the analysis in Section 4 accordingly. Importantly, the overall conclusions of the paper remain unchanged. **With the corrected setup, the Eurus in-domain gain is 15.32% larger than the out-of-domain gain, and the average ID–OOD difference across models increases to 6.1%, as reported in the revised Table 2.**
>
> We further examined every remaining RPT model whose in-domain improvement does not exceed its out-of-domain improvement. Our analysis shows that these cases can all be explained by limitations in the RPT training data:
>
> - Model (5), Absolute_Zero_Reasoner-3B, was trained on a very small amount of code data. This data does not capture the diversity of tasks in the evaluation suite, which includes multiple programming languages and a wide range of difficulty levels. As a result, the model does not learn the full structure of the in-domain distribution, and the in-domain improvement becomes weaker.
> - Models (8) Llama-3.1-Nemotron-Nano-8B-v1 and (9) Thespis-Llama-3.1-8B were trained on general instruction following and chat data. These datasets do not reflect the variability of the knowledge-intensive domains used in our evaluation, which explains why their out-of-domain gains appear comparable or larger.
> - Models (13) OREAL-7B and (14) Open-RS3 RPT were trained mainly on simple math tasks. The evaluation tasks cover substantially more challenging math problems, so the improvement within the math domain is limited.
>
> To isolate data effects from implementation effects, we also empirically analyzed a variety of RPT algorithms, model sizes, backbone architectures, and training steps, as described in Section 4.4. Across all of these implementation variations, the pattern is consistent: in-domain improvements are always larger than out-of-domain improvements. These results directly support our earlier analysis that the variance in RPT generalizability arises more from dataset-related factors than from implementation configurations or model architecture.
>
> Finally, to directly test whether broader domain coverage improves generalizability, we trained an additional model on a mixed domain dataset. We created this dataset by randomly sampling 13,333 examples from each single domain dataset. As we report in Appendix E, Table 5, the mixed-domain RPT model achieves an overall 2.7% accuracy improvement compared to single-domain RPT models. This confirms that the breadth and structure of the RPT training data play a central role in determining generalizability.

---

> ### Author Response · Authors · 2025-11-21
> **Response to Reviewer FGBZ (2/2): W2, W3, W4**
>
> **W2 (heterogeneity of knowledge-intensive tasks):** We do not view this as a domain mismatch but rather as an inherent characteristic of knowledge-intensive tasks, which are highly diverse and context-dependent. Consequently, even within the same nominal domain, performance variance can be large due to differences in factual grounding and question styles. This is further supported quantitatively by the large divergence among knowledge-intensive evaluation tasks, as we show in Appendix B.2.
>
> **W3 (quantifying reasoning template hypothesis):** To more thoroughly evaluate the hypothesis in Sections 2 and 4 on reasoning templates, we perform a quantitative analysis of reasoning–step distributions across and within domains (Appendix B).
>
> We begin by randomly sampling $\min(\text{dataset size}, 100)$ tasks from each evaluation benchmark, yielding 1,470 task instances in total. We use Claude Sonnet 4.5, the state-of-the-art periphery reasoning model that exposes reasoning traces, to complete these tasks. We then collect the generated reasoning traces and use GPT-4o to tag each reasoning step according to the taxonomy defined in Appendix B. These tagged traces allow us to quantitatively compare the reasoning templates used across domains. We present the detailed distributions in Figure 6 in Appendix B. To measure similarity, we use the Jeffreys divergence [1], a symmetric divergence between probability distributions. We have the following statistics:
>
> - *Across-domain reasoning templates’ similarity (Appendix B.1, Section 4.2 lines 401-407):* We compute the Jeffreys divergence between domain-level reasoning–step distributions by aggregating and normalizing all tagged traces within each domain. Using these aggregated distributions, the Jeffreys divergence between math and code is 0.18, between math and knowledge-intensive is 0.29, and between code and knowledge-intensive is 0.69. These values confirm the hypothesis that math and code rely on more similar reasoning templates, whereas knowledge-intensive tasks require substantially different reasoning processes.
>
> - *Intra-domain reasoning templates’ similarity  (Appendix B.2, Section 4.3, lines 441-465):* To assess how diverse each domain is internally, we compute the Jeffreys divergence between every pair of datasets within the same domain, using each dataset’s normalized reasoning–step distribution. Averaging across all pairwise comparisons, the mean Jeffreys divergence within math is 0.15, within code is 0.14, and within knowledge-intensive tasks is 0.19. This indicates that knowledge-intensive tasks exhibit more heterogeneous reasoning templates, while math and code datasets are more internally consistent.
>
> Together, these results provide direct quantitative evidence that **(1)** the reasoning templates in math and code are indeed more similar to each other than to those in the knowledge-intensive domain, and **(2)** the reasoning templates within the knowledge-intensive domain are indeed more diverse.
>
> **W4 (different backbone models):** Beyond the DeepSeek-R1-Distill-Qwen-1.5B backbone model used in our main experiments, we additionally evaluate a different architecture, Llama-3.2-3B-Instruct, and observe consistent trends:  the in-domain vs. out-of-domain accuracy improvement gap is nearly identical (4.9% for DeepSeek-R1-Distill-Qwen-1.5B vs. 5.1% for Llama-3.2-3B-Instruct), and the ratio of in-domain to out-of-domain odds is similarly close (1.28 for DeepSeek-R1-Distill-Qwen-1.5B vs. 1.38 for Llama-3.2-3B-Instruct). This suggests that the lack of generalizability in RPT is inherent to the RPT process itself, rather than a consequence of the base model architecture or pretraining data. We report the full numerical results in Table 3 of the revised paper and provide a detailed interpretation in Section 4.4 (lines 474-479).
>
> [1] Harold Jeffreys. Theory of Probability. Oxford University Press, Oxford, UK, 3 edition, 1961

---

> > ### Comment · Reviewer_FGBZ · 2025-11-21
> >
> > Thank you so much for the detailed response! It's great to see that you updated the weird prime results and this makes the paper much stronger. Additionally, I appreciate all the additional experiments around reasoning templates. I think all my core questions have been answered and am happy to raise my score :)

---

> ### Author Response · Authors · 2025-11-22
> **Thank you!**
>
> Dear Reviewer FGBZ,
>
> Thank you for your support and for raising the score! We really appreciate the time you took to revisit our work. Your advice genuinely helped us improve the paper — especially the insights that guided us to investigate and update the outlier results, as well as expand the reasoning-template experiments to better ground our explanation of RPT generalizability. These changes have strengthened our paper substantially, and we are very grateful for your thoughtful and constructive feedback.
>
> Regards,
>
> Paper 14011 Authors

---

### Official Review · Reviewer_8H9o · 2025-10-31

**Soundness:** 3
**Presentation:** 3
**Contribution:** 3
**Rating:** 8
**Confidence:** 4

**Summary:**

This paper conducts an extensive study on publicly released models to understand the cross-domain skill transfer during reinforcement learning. They explore mathematical, code, and knowledge-intensive reasoning, evaluating how much performance improves from the base model when models are trained on data from different domains.

**Strengths:**

The results, while not incredibly surprising for those with substantial experience performing RL finetuning on language models, are quite valuable to see. The study is quite broad, only models with publicly available training data are included, and the experimental design is sound.

**Weaknesses:**

It would be helpful to list the models that you tested, both for reproducibility and for clarity. One question I have is how diverse the *base* model pool was; e.g. were most models based on Qwen (which is quite strong on math and code already), or was there a diverse set of model families included in your study?

If possible, it would be very enlightening if there could be a further study on the *kinds* of reasoning each model uses, to see if there are explicit strategies common amongst them (so we can better understand what "tools" models need for e.g. math or code), but this is mostly out of scope of this paper.

**Questions:**

When selecting domains, did you consider any others? If time allows, I think instruction following is a good verifiable domain to explore as well, and previous work has shown that models struggle to generalize to constraints beyond those they were trained on: https://arxiv.org/abs/2507.02833

Also, I'd recommend tweaking the citations in the first paragraph, right now there's essentially just a run on sentence of citations.

---

> ### Author Response · Authors · 2025-11-21
> **Response to Reviewer 8H9o (1/2): W1, W2**
>
> Dear Reviewer 8H9o,
>
> Thank you for your detailed summary and constructive feedback. We are glad that you find our work valuable, broad, and sound. We address your comments as follows.
>
> **W1 (base model pool):** We have documented all evaluated models in Table 1, including both the base and RFT models. Our base model pool is intentionally diverse: architecturally, it includes Llama- and Qwen-based models, and in terms of pretraining data, it spans models trained on widely different corpora. This diversity allows us to more reliably measure RPT generalizability across heterogeneous model families rather than a single lineage.
>
> **W2 (reasoning strategies across domains):** To better understand what reasoning strategies are required for different domains to inspire the improvement of generalizability of RPT, we perform a quantitative analysis of reasoning-step distributions across and within domains (Appendix B).
>
> We begin by randomly sampling $\min(\text{dataset size}, 100)$ tasks from each evaluation benchmark, yielding 1,470 task instances in total. We use Claude Sonnet 4.5, the state-of-the-art periphery reasoning model with exposed reasoning traces, to complete these tasks. We then collect the generated reasoning traces and use GPT-4o to tag each reasoning step according to the taxonomy defined in Appendix B. These tagged traces allow us to quantitatively compare the reasoning templates used across domains.
>
> We present the detailed distributions in Figure 6 in Appendix B. We find that EXECUTE_STEP is the most dominant category across all three domains. PLAN is particularly important in math and code tasks, reflecting their structured, multi-step problem-solving nature, while SETUP plays a comparatively larger role in knowledge-intensive tasks, where establishing context or recalling background information is crucial. These findings suggest that enhancing RPT with domain-specific reasoning tools, such as stronger planning operators for math and code, and richer retrieval mechanisms for knowledge tasks, may further improve its generalizability.

---

> ### Author Response · Authors · 2025-11-21
> **Response to Reviewer 8H9o (2/2): Q1, Q2**
>
> **Q1 (Instruction-Following as an additional domain):** Thank you for the helpful suggestion and for pointing us to the related work. We included Model Pair 8 in our original submission, which was trained solely on the Instruction Following domain. We initially classified this data under the knowledge-intensive domain, but after reviewing your recommended paper [1], we agree that treating Instruction Following as its own domain provides finer granularity.
>
> To make our analysis more complete, we additionally conduct an interventional evaluation on IFBench [1], the paper you recommended, and IFEval [2], a widely used benchmark measuring instruction-following capability. Below, we report the raw scores for the base model and the Math-RPT, Code-RPT, and Knowledge-RPT models evaluated on the two Instruction Following benchmarks:
>
> ### **Performance on Instruction-Following Benchmarks**
> | Model               | IFEval Accuracy | IFBench Accuracy |
> |---------------------|:---------------:|:----------------:|
> | Base Model          |      0.377      |      0.144       |
> | Math-RPT Model      |      0.375      |      0.138       |
> | Code-RPT Model      |      0.354      |      0.137       |
> | Knowledge-RPT Model |      0.382      |      0.140       |
>
>
> ### **Domain-wise Improvement (Augmenting Figure 4(a))**
> | RPT Model           | Instruction-Following |   Math   |   Code   | Knowledge |
> |---------------------|:---------------------:|:--------:|:--------:|:---------:|
> | Math-RPT Model      |        −0.34%         | +1.24%   | +1.70%   | −1.93%    |
> | Code-RPT Model      |        −1.73%         | −0.15%   | +4.21%   | −3.98%    |
> | Knowledge-RPT Model |        +0.02%         | +1.17%   | −0.02%   | −3.10%    |
>
>
> ### **Domain-wise Odds Ratio (Augmenting Figure 4(b))**
> | RPT Model           | Instruction-Following |  Math   |  Code   | Knowledge |
> |---------------------|:---------------------:|:-------:|:-------:|:---------:|
> | Math-RPT Model      |         0.91*         |  1.24*  |  1.15*  |   0.92*   |
> | Code-RPT Model      |         0.98          |  0.98   |  1.38*  |   0.96*   |
> | Knowledge-RPT Model |         1.00          |  1.08*  |  1.00   |   0.88*   |
>
>
>
> We find that introducing Instruction Following as a separate domain strengthens our earlier conclusions that RPT models exhibit limited generalizability outside their training domains:
> - *Math-RPT and Code-RPT models perform worse on instruction-following tasks:* Instruction-following prompts tend to be open-ended, unstructured, and diverse, so the highly specialized reasoning templates learned during math and code RPT do not transfer.
> - *Knowledge-RPT models perform only slightly better:* While knowledge-intensive tasks (e.g., finance, medical) are closer in style, they are still domain-specific, whereas instruction following involves everyday reasoning, such as “Which brand of sneakers is better: Prada or Nike?” This mismatch limits generalization, though the degradation is smaller compared to math or code RPT.
>
> From these findings, we believe an important direction for future work is to refine our current domain taxonomy into more fine-grained subdomains: for example, separating instruction following as you suggested, distinguishing engineering-style tasks within the knowledge domain [3], and/or grouping datasets by programming language within the code domain. A deeper understanding of these finer-grained task families will help reveal what types of reasoning RPT actually transfers, thereby guiding the development of more robust and generalizable RPT methods. We view our work as a foundation for this line of investigation.
>
> [1] Pyatkin, Valentina, et al. "Generalizing Verifiable Instruction Following." arXiv preprint arXiv:2507.02833 (2025).
>
> [2] Zhou, Jeffrey, et al. "Instruction-following evaluation for large language models." arXiv preprint arXiv:2311.07911 (2023).
>
> [3] Guo, Xingang, et al. "Toward Engineering AGI: Benchmarking the Engineering Design Capabilities of LLMs." arXiv preprint arXiv:2509.16204 (2025).
>
> **Q2 (Introduction readability and citation structure):** We revised the introduction to improve readability by restructuring the citations, grouping related work together, and removing long uninterrupted citation lists (Section 1, lines 27–38).

---

> ### Author Response · Authors · 2025-12-03
> **Summary of Responses to Reviewer 8H9o**
>
> Dear Reviewer 8H9o,
>
> Thank you again for your thoughtful comments and constructive feedback. Since the discussion phase is approaching its end, we would like to summarize that **we have fully addressed all of your concerns**. Specifically:
>
> - **W1:** We documented all base and RPT models, including their RPT data, selection criteria, etc., in Section 3 (Table 1), confirming diversity across architectures and pretraining corpora.
>
> - **W2:** We conducted a quantitative cross-domain reasoning step analysis in Appendix B (Figure 6), demonstrating different reasoning patterns across domains. These patterns provide insights into what domain-specific reasoning tools may be effective in improving generalizability.
>
> - **Q1:** We added Instruction Following as a separate domain and evaluated RPT models on IFBench and IFEval. The results in [our response](https://openreview.net/forum?id=mvLhN0veUd&noteId=aOHrI2nUis) strengthen our conclusion that RPT models show limited generalizability outside their training domains.
>
> - **Q2:** We revised our introduction to improve readability by reorganizing the narrative and grouping citations more coherently (Section 1, lines 27–38).
>
> Regards,
>
> Paper 14011 Authors

---

### Official Review · Reviewer_TWrh · 2025-11-01

**Soundness:** 3
**Presentation:** 3
**Contribution:** 3
**Rating:** 6
**Confidence:** 4

**Summary:**

The paper examines whether reasoning gains from RPT generalize beyond the training domains. Through both observational and controlled interventional studies across math, code, and knowledge-intensive tasks, the authors find that RPT improvements are domain-specific, effective within similar structured domains, eg. math and code, but failing to transfer to unstructured ones, e.g., legal, medical. The work highlights the limited cross-domain generalizability of current RPT approaches.

**Strengths:**

- The paper tackles an important and timely question about whether reasoning improvements from reinforcement post-training can truly generalize beyond the training domain.
- The study design is comprehensive and convincing, combining large-scale observational analysis of public RPT models with controlled interventional experiments under unified settings.
- The experiments are extensive and well-documented, covering 16 diverse benchmarks across mathematics, code, and knowledge reasoning with appropriate statistical validation.

**Weaknesses:**

- The experiments are conducted on relatively small models (up to 8B) with limited-scale RPT training, leaving it unclear whether the same generalization patterns would persist under larger LLMs.
- The paper stops short of analyzing how different aspects of RPT training, such as reward signal quality or optimization dynamics, might contribute to the observed lack of cross-domain transfer, leaving the underlying cause somewhat underexplored.
- The paper does not include any longitudinal or ablation analysis during training, which could reveal how generalization patterns evolve over time or collapse across domains.
- The interventional experiments are all based on a single backbone DeepSeek-R1-Distill-Qwen-1.5B), so the conclusions are lacking in generality as the observed trends may depend on that model’s pre-training distribution.

**Questions:**

- In the interventional experiments, were the three single-domain RPT models trained with identical reward functions or domain-specific ones? Clarifying such details could help interpret whether the observed generalization gaps stem from reward differences or reasoning differences.
- Could the authors comment on whether a mixed-domain RPT training setting, e.g., combining math, code, and knowledge reasoning, might mitigate the observed specialization? This would help verify whether domain isolation itself causes the loss of generalization.
- Have the authors considered analyzing intermediate checkpoints during RPT training to see if cross-domain performance degrades gradually or abruptly? Such temporal analysis might shed light on when specialization emerges.

---

> ### Author Response · Authors · 2025-11-21
> **Response to Reviewer TWrh (1/3): W1**
>
> Dear Reviewer TWrh,
>
> Thank you for your detailed summary and constructive feedback. We are glad that you find our work tackling an important and timely problem through a detailed and systematic design. We address your comments as follows.
>
> **W1 (limitation of small model sizes):**  First, we want to stress that our focus on small-scale models aligns with the established research practice: the most influential academic work in RPT primarily trains and evaluates small-scale models. Representative examples include:
>
> - DPO [1], cited 6824 times as of 11/19/2025, largest model size 6.9B
> - RAFT [2], cited 593 times as of 11/19/2025, largest model size 7B
> - Search-R1 [3], cited 418 times as of 11/19/2025, starred 3.5k, within 8 months of release, largest model size 7B
> - Eurus [4], cited 175 times as of 11/19/2025, starred 1.8k, within 9 months of release, largest model size 7B
> - Absolute Zero [5], cited 74 times as of 11/19/2025, starred 1.7k, within 6 months of release, largest model size 14B (main experiment largest size 7B)
>
> These works have been widely adopted and heavily cited, and their conclusions have not been regarded as invalid or non-generalizable solely because they use small models. On the contrary, the field has treated small-model RPT as a scientifically meaningful and practically relevant setting, not only due to computational constraints but also because the core philosophy of RPT is to use small curated datasets to improve small models on complex reasoning tasks. Therefore, evaluating the generalizability of small models is valid and important, and provides sufficient evidence for understanding RPT behavior.
>
> To further support this, 10 out of the originally selected 14 RPT model pairs already involve the largest publicly available variant within their respective model families. For the remaining four models, **we additionally evaluate their largest released variants** (DeepCoder-14B, OREAL-32B, Fin-o1-14B, and AZR-Coder-14B) and compare them against their smaller variants in the revised paper. We find that larger models exhibit comparable or even worse generalizability: out-of-domain accuracy improvement is 16.5% less than in-domain accuracy improvement as model size increases, and the ratio of in-domain to out-of-domain odds increases by 6.6×. This degradation arises because larger models more strongly overfit to the domain on which they are RPT-trained. These results reinforce our main conclusion: RPT gains do not generalize. Although this finding is largely derived from small models, it holds for large models both theoretically and empirically. We report the full numerical results in Appendix C, Table 4 of the revised paper, and provide a detailed interpretation in Section 4.4 (lines 504–509).
>
> [1] Rafailov, Rafael, et al. "Direct preference optimization: Your language model is secretly a reward model." Advances in neural information processing systems 36 (2023): 53728-53741.
>
> [2] Dong, Hanze, et al. "Raft: Reward ranked finetuning for generative foundation model alignment." arXiv preprint arXiv:2304.06767 (2023).
>
> [3] Jin, Bowen, et al. "Search-r1: Training llms to reason and leverage search engines with reinforcement learning." arXiv preprint arXiv:2503.09516 (2025).
>
> [4] Cui, Ganqu, et al. "Process reinforcement through implicit rewards." arXiv preprint arXiv:2502.01456 (2025).
>
> [5] Zhao, Andrew, et al. "Absolute zero: Reinforced self-play reasoning with zero data." arXiv preprint arXiv:2505.03335 (2025).

---

> ### Author Response · Authors · 2025-11-21
> **Response to Reviewer TWrh (2/3): W2, W3, Q3, W4**
>
> **W2 (reward quality and optimization dynamics):** First, we want to clarify that in our experiments, we followed the best practices in RPT that have been widely adopted and shown to achieve state-of-the-art results in prior work [1, 2, 3]. We now provide additional quantitative analyses on different aspects of RPT training:
>
> - ***Reward signals are consistently accurate and high quality for all domains:*** For math, we used both the string-based and symbolic equivalence checking via the math-verify package [4]. For code, we used unit tests. For general knowledge, we used the string-based equivalence checking. To quantitatively assess the reward quality, we sampled 100 training examples and manually verified the correctness of reward signals. We find that the correctness rates of math, code, and general knowledge rewards are 94%, 99%, and 97%, respectively, indicating similarly high reward quality across domains. We clarify this point in Appendix D.1 of the revised paper.
>
> - ***Optimization dynamics show steady and valid model learning:*** We added an analysis of optimization behavior in the revision (Appendix D.2), examining both training reward and entropy over the course of RPT training. Across domains, we observe stable and consistent dynamics. We find that all runs exhibit increasing training rewards, with no signs of pathological optimization behavior. Quantitatively, over the one-epoch (625-step) training, the 50-step average training reward of math, code, and knowledge domains increases by 9.7%, 21.7%, and 71.0%, respectively (Figure 7).
>
> These quantitative results show that our training configuration is sound and methodologically well-grounded for developing rigorous conclusions on RPT generalizability.
>
> [1] Luo, Michael, et al. “DeepScaleR: Surpassing O1-Preview with a 1.5B Model by Scaling RL”
>
> [2] Luo, Michael, et al. “DeepCoder: A Fully Open-Source 14B Coder at O3-mini Level”
>
> [3] Yu, Qiying, et al. "Dapo: An open-source llm reinforcement learning system at scale." arXiv preprint arXiv:2503.14476 (2025).
>
> [4] Hynek Kydlíček and Greg Gandenberger. Math-verify: A robust mathematical expression evaluation system, 2025.
>
> **W3, Q3 (checkpoint evaluations):** We conduct additional experiments that double the number of RPT training steps, and evaluate intermediate checkpoints to track how generalization evolves throughout training. As we show in Figure 5, the gap between in-domain and out-of-domain performance gains steadily increases and eventually converges. Specifically, the gap after the second epoch increases by 13.6% relative to the first epoch. These results align with our intuition: as the model is more exposed to domain-specific data, it becomes increasingly specialized and thus more prone to overfitting and lower generalizability. We provide a detailed interpretation in Section 4.4, lines 480-485.
>
> **W4 (different backbone models):** Beyond the DeepSeek-R1-Distill-Qwen-1.5B backbone model used in our main interventional analysis, we additionally evaluate a different architecture, Llama-3.2-3B-Instruct, and observe consistent trends:  the in-domain vs. out-of-domain accuracy improvement gap is nearly identical (4.9% for DeepSeek-R1-Distill-Qwen-1.5B vs. 5.1% for Llama-3.2-3B-Instruct), and the ratio of in-domain to out-of-domain odds is similarly close (1.28 for DeepSeek-R1-Distill-Qwen-1.5B vs. 1.38 for Llama-3.2-3B-Instruct). This suggests that the lack of generalizability in RPT is inherent to the RPT process itself, rather than a consequence of the base model architecture or pretraining data. We report the full numerical results in Table 3 of the revised paper and provide a detailed interpretation in Section 4.4 (lines 474-479).

---

> ### Author Response · Authors · 2025-11-21
> **Response to Reviewer TWrh (3/3): Q1, Q2**
>
> **Q1 (reward function clarification):** We clarify in Appendix D.1 of the revised paper that following prior work that has demonstrated promising performance [1,2,3], we use domain-specific, binary reward functions:
>
> - Math: reward = 1 if the model’s answer is mathematically equivalent to the ground truth; otherwise, 0.
> - Code: reward = 1 if the model’s answer passes all unit tests; otherwise, 0.
> - Knowledge: reward = 1 if the model’s answer matches the ground truth string; otherwise, 0.
>
> Because the reward structure is identical (binary reward functions) and the correctness checks are standard and reliable across domains, this confirms that the lack of RPT generalizability indeed stems from differences in reasoning rather than reward signal discrepancies across domains.
>
> **Q2 (RPT on mixed domain data):** To verify the effect of domain isolation, we train an additional model using a mixed-domain dataset, created by randomly sampling 13,333 data points from each of the single-domain datasets. As we report in Appendix E, Table 5, the mixed-domain RPT model achieves an overall 2.7% accuracy improvement compared to single-domain RPT models.
>
> Since the mixed-domain dataset was constructed by randomly sampling from each domain, we expect its performance to be suboptimal. With more optimized cross-domain data curation strategies, as demonstrated in prior work [5, 6], mixed-domain RPT could potentially achieve even stronger gains. This observation further strengthens our conclusion that RPT, when trained on narrow or single-domain data, lacks generalizability to unseen domains, and that broader domain exposure alleviates this issue.
>
> [5] Guo, Daya, et al. "Deepseek-r1: Incentivizing reasoning capability in llms via reinforcement learning." arXiv preprint arXiv:2501.12948 (2025).
>
> [6] Team, Kimi, et al. "Kimi k2: Open agentic intelligence." arXiv preprint arXiv:2507.20534 (2025).

---

> ### Author Response · Authors · 2025-12-03
> **Summary of Responses to Reviewer TWrh**
>
> Dear Reviewer TWrh,
>
> Thank you again for your thoughtful comments and constructive feedback. Since the discussion phase is approaching its end, we would like to summarize that **we have fully addressed all of your concerns**. Specifically:
>
> - **W1:** We additionally evaluated the largest variants for all model pairs in our observational study. Results in Section 4.4 (lines 504-509) and Appendix C (Table 4) show that RPT’s lack of generalizability **remains consistent for larger models**.
>
> - **W2:** We added quantitative analyses of reward correctness (Appendix D.1, 94–99% across domains) and optimization behavior (Appendix D.2; Figure 7) in our interventional study. Results show **sound reward signals and stable dynamics**, further confirming that RPT’s lack of generalizability arises from their fundamental limitations, rather than training configuration issues.
>
> - **W3, Q3:** We doubled RPT training steps in our interventional study and evaluated 12 intermediate checkpoints. Results in Section 4.4 (Figure 5, lines 480–485) and Appendix H **show as training progresses, the ID–OOD gap first grows and then gradually converges after the first full epoch**, once the model has seen all data. This temporal trajectory reveals a clear pattern of RPT specialization into the training domain.
>
> - **W4:** We additionally conducted our interventional study with Llama-3.2-3B-Instruct. Results in Section 4.4 (Table 3, lines 474–479) show ID–OOD gaps nearly identical to our original setup with DeepSeek-R1-Distill-Qwen-1.5B, i.e., **difference in the base models, including architecture, pretraining data distributions, etc. does not change the generalization behavior**.
>
> - **Q1:** We clarify in Appendix D.1 (lines 1023-1052) that we apply the same binary reward functions for all domains, with domain-specific correctness checks following best practice, **confirming that the lack of RPT generalizability is rooted in reasoning differences** rather than surface-level reward signal differences across domains.
>
> - **Q2:** We additionally trained a mixed-domain RPT model. Results in Appendix E (Table 5) show a 2.7% improvement over single-domain RPTs, **confirming that domain isolation contributes to RPT's lack of generalizability**.
>
> Regards,
>
> Paper 14011 Authors

---

### Official Review · Reviewer_172Y · 2025-11-01

**Soundness:** 2
**Presentation:** 3
**Contribution:** 2
**Rating:** 4
**Confidence:** 4

**Summary:**

This paper presents an empirical study that assess to which degree reinforcement learning post-training enables generalizable improvement of reasoning across domains. It is structured into two parts, an observational and an interventional study.

**Strengths:**

The topic addresses a currently open important question for reasoning LLMs.

Certainly a strength of the paper is its systematic and transparent setup (in particular for the selection of tested models) and statistical evaluation.

**Weaknesses:**

A key weakness of the study is the focus on small models. While I understand the computational limitations. However, it seems reasonable that a certain model complexity might be required to actually generalize across domains. Therefore, it is not clear how the findings actually generalize to larger models that might in any case better suited for complex reasoning tasks.

Similarly, only one particular Reinforcement Learning process is tested for fine tuning with a single snapshot after one epoch. Here, it would be key to also see the development over multiple snapshots. With the current setup, one could hypothesize that generalization just sets in later. An evaluation would be interesting.

The paper overall is very sparse with the exact evaluation results for the different tests and models. I would expect that the paper reports on the detailed per task per model accuracies.

The used evaluation measure is fine. However, I think there is a large difference between an increase of accuracy from 60% to 61% or an improvement from 95% to 96%. In other words, the relative improvement is also important and should be addressed in a second measure.

**Questions:**

The issues to be discussed in my opinion can be derived straightforward from the weaknesses section.

---

> ### Author Response · Authors · 2025-11-21
> **Response to Reviewer 172Y (1/3): W1**
>
> Dear Reviewer 172Y,
>
> Thank you for your detailed summary and constructive feedback. We are glad that you find our work addressing an important problem through systematic and transparent setups. Below, we address the mentioned weaknesses.
>
> **W1 (limitation of small model sizes):** First, we want to stress that our focus on small-scale models aligns with the established research practice: the most influential academic work in RPT primarily trains and evaluates small-scale models. Representative examples include:
>
> - DPO [1], cited 6824 times as of 11/19/2025, largest model size 6.9B
> - RAFT [2], cited 593 times as of 11/19/2025, largest model size 7B
> - Search-R1 [3], cited 418 times as of 11/19/2025, starred 3.5k, within 8 months of release, largest model size 7B
> - Eurus [4], cited 175 times as of 11/19/2025, starred 1.8k, within 9 months of release, largest model size 7B
> - Absolute Zero [5], cited 74 times as of 11/19/2025, starred 1.7k, within 6 months of release, largest model size 14B (main experiment largest size 7B)
>
> These works have been widely adopted and heavily cited, and their conclusions have not been regarded as invalid or non-generalizable solely because they use small models. On the contrary, the field has treated small-model RPT as a scientifically meaningful and practically relevant setting, not only due to computational constraints but also because the core philosophy of RPT is to use small curated datasets to improve small models on complex reasoning tasks. Therefore, evaluating the generalizability of small models is valid and important, and provides sufficient evidence for understanding RPT behavior.
>
> To further support this, 10 out of the originally selected 14 RPT model pairs already involve the largest publicly available variant within their respective model families. For the remaining four models, **we additionally evaluate their largest released variants** (DeepCoder-14B, OREAL-32B, Fin-o1-14B, and AZR-Coder-14B) and compare them against their smaller variants in the revised paper. We find that larger models exhibit comparable or even worse generalizability: out-of-domain accuracy improvement is 16.5% less than in-domain accuracy improvement as model size increases, and the ratio of in-domain to out-of-domain odds increases by 6.6×. This degradation arises because larger models more strongly overfit to the domain on which they are RPT-trained. These results reinforce our main conclusion: RPT gains do not generalize. Although this finding is largely derived from small models, it holds for large models both theoretically and empirically. We report the full numerical results in Appendix C, Table 4 of the revised paper, and provide a detailed interpretation in Section 4.4 (lines 504–509).
>
> [1] Rafailov, Rafael, et al. "Direct preference optimization: Your language model is secretly a reward model." Advances in neural information processing systems 36 (2023): 53728-53741.
>
> [2] Dong, Hanze, et al. "Raft: Reward ranked finetuning for generative foundation model alignment." arXiv preprint arXiv:2304.06767 (2023).
>
> [3] Jin, Bowen, et al. "Search-r1: Training llms to reason and leverage search engines with reinforcement learning." arXiv preprint arXiv:2503.09516 (2025).
>
> [4] Cui, Ganqu, et al. "Process reinforcement through implicit rewards." arXiv preprint arXiv:2502.01456 (2025).
>
> [5] Zhao, Andrew, et al. "Absolute zero: Reinforced self-play reasoning with zero data." arXiv preprint arXiv:2505.03335 (2025).

---

> ### Author Response · Authors · 2025-11-21
> **Response to Reviewer 172Y (2/3): W2**
>
> **W2 (lack of RPT configuration variants):** First, we would like to stress that our observational study spans 18 pairs of RPT models, covering a broad range of algorithmic, architectural, and implementation choices. To systematically capture these variants at a finer granularity, we have additionally included the following experiments in our interventional analysis (full setup in Section 3, lines 267–281):
>
> - *(Training-step variations)* We conduct additional experiments that double the number of RPT training steps, and evaluate intermediate checkpoints to track how generalization evolves throughout training. As we show in Figure 5, the gap between in-domain and out-of-domain performance gains steadily increases and eventually converges. Specifically, the gap after the second epoch increases by 13.6% relative to the first epoch. These results indicate that the lack of generalizability in RPT is not resolved by simply increasing the amount of training. This aligns with our intuition: as the model is more exposed to domain-specific data, it becomes increasingly specialized and thus more prone to overfitting. We provide a detailed interpretation in Section 4.4, lines 480-485.
>
> - *(RPT algorithm variations)* In addition to the GRPO algorithm used in our main interventional analysis, we evaluate an alternative RPT algorithm, DAPO [6]. The results remain consistent: the in-domain vs. out-of-domain accuracy improvement gap is nearly identical (4.9% for GRPO vs. 5.2% for DAPO), and the ratio of in-domain to out-of-domain odds is similarly close (1.28 for GRPO vs. 1.31 for DAPO). This suggests that despite the procedural differences of different RPT algorithms, the core optimization behavior of RPT dominates. We report the full numerical results in Table 3 of the revised paper and provide a detailed interpretation in Section 4.4 (lines 469–473).
>
> - *(Base model variations)* Beyond the DeepSeek-R1-Distill-Qwen-1.5B backbone model used in our main interventional analysis, we additionally evaluate a different architecture, Llama-3.2-3B-Instruct, and observe consistent trends:  the in-domain vs. out-of-domain accuracy improvement gap is nearly identical (4.9% for DeepSeek-R1-Distill-Qwen-1.5B vs. 5.1% for Llama-3.2-3B-Instruct), and the ratio of in-domain to out-of-domain odds is similarly close (1.28 for DeepSeek-R1-Distill-Qwen-1.5B vs. 1.38 for Llama-3.2-3B-Instruct). This suggests that the lack of generalizability in RPT is inherent to the RPT process itself, rather than a consequence of the base model architecture or pretraining data. We report the full numerical results in Table 3 of the revised paper and provide a detailed interpretation in Section 4.4 (lines 474-479).
>
> These expanded experiments further strengthen our main conclusion by demonstrating that it is robust to a wide range of RPT setups.
>
> [6] Yu, Qiying, et al. "Dapo: An open-source llm reinforcement learning system at scale." arXiv preprint arXiv:2503.14476 (2025).

---

> ### Author Response · Authors · 2025-11-21
> **Response to Reviewer 172Y (3/3): W3, W4**
>
> **W3 (per-task, per-model accuracy):** We include detailed per-task, per-model accuracy results in Appendix F for greater transparency. Specifically, we include:
>
> - The full observational study results of 36 models * 16 tasks in Appendix F.1, Table 6.
> - The full interventional study results of 12 models * 16 tasks in Appendix F.2, Table 7.
>
> **W4 (relative improvements):** For completeness, we additionally report *relative improvement* as a complementary metric for all main-paper results in Appendix G. This complementary metric strengthens our claims from the following three dimensions:
> - Domain-wise performance gain gaps become substantially more pronounced: The absolute in-domain vs. out-of-domain performance gap in Table 2 is 6.07%, but the corresponding relative improvement gap in Table 8 is 256.71%, meaning the in-domain improvements are 3.5× larger. The same pattern holds across all three remaining analyses using the relative improvement metric (Figures 8–10).
>
> - Our chosen configuration for the interventional study (Qwen + GRPO) is clearly the strongest: As shown in Table 9, Llama + GRPO exhibits a relative performance gap of 28.38%, while Qwen + GRPO has only 8.75%. In contrast, their absolute accuracy gaps in Table 3 are very similar (5.23% vs. 4.94%). This relative metric therefore reveals differences that the absolute metric obscures.
>
> - The growth and eventual convergence of in-domain vs. out-of-domain gains is more visible: in Figure 11, the rise and subsequent stabilization of the relative improvement gap over training epochs is noticeably clearer than when using the absolute metrics shown in Figure 5.

---

> ### Author Response · Authors · 2025-12-03
> **Summary of Responses to Reviewer 172Y**
>
> Dear Reviewer 172Y,
>
> Thank you again for your thoughtful comments and constructive feedback. Since the discussion phase is approaching its end, we would like to summarize that **we have fully addressed all of your concerns**. Specifically:
>
> - **W1:** We additionally evaluated the largest variants for all model pairs in our observational study. Results in Section 4.4 (lines 504-509) and Appendix C (Table 4) show that **larger models do not generalize better**.
>
> - **W2.1:** We doubled RPT training steps in our interventional study and evaluated 12 intermediate checkpoints. Results in Section 4.4 (Figure 5, lines 480–485) and Appendix H show that the ID–OOD gap grows as training progresses, i.e., **more RPT training steps does not improve generalizability**.
>
> - **W2.2:** We additionally conducted our interventional study with DAPO. Results in Section 4.4 (Table 3, lines 469–473) show ID–OOD gaps nearly identical to our original setup with GRPO, i.e., **RPT's lack of generalizability is consistent across different algorithms**.
>
> - **W2.3:** We additionally conducted our interventional study with Llama-3.2-3B-Instruct. Results in Section 4.4 (Table 3, lines 474–479) show ID–OOD gaps nearly identical to our original setup with DeepSeek-R1-Distill-Qwen-1.5B, i.e., **RPT's lack of generalizability is consistent across different base models**.
>
> - **W3:** We included the full per-task, per-model results for all evaluations in Appendix F.1 Table 6 (observational study) and Appendix F.2 Table 7 (interventional study, including all configuration variants).
>
> - **W4:** We added relative improvement analysis in Appendix G for all evaluations, which significantly strengthens consistent conclusions across all settings.
>
> Regards,
>
> Paper 14011 Authors

---

### Author Response · Authors · 2025-12-03
**Summary of Reviewer Consensus and Our Responses for Paper 14011**

Dear Area Chairs,

Thank you for coordinating the review process and handling our submission, and we thank all reviewers for their valuable feedback! We are encouraged that all reviewers (Reviewers 172Y, TWrh, 8H9o, FGBZ) recognize **(1)** the importance of the problem we study, **(2)** the transparency and rigor of our setup and design, and **(3)** the breadth and comprehensiveness of our evaluations.

We are grateful to [Reviewer FGBZ for recognizing that our response fully addressed all of their concerns](https://openreview.net/forum?id=mvLhN0veUd&noteId=M1ZDdeaBsT) and **for raising their score from 6 to 8 on Nov 21st**, within a day after our initial rebuttal. We believe that we have also **fully addressed all the concerns raised by the remaining 3 reviewers,** who did not reply before the changes took place, in our individual rebuttal responses, and we have additionally summarized them in our final responses to [Reviewer 172Y](https://openreview.net/forum?id=mvLhN0veUd&noteId=WnfmQSyEIC), [Reviewer TWrh](https://openreview.net/forum?id=mvLhN0veUd&noteId=aVanwG7f2y), and [Reviewer 8H9o](https://openreview.net/forum?id=mvLhN0veUd&noteId=wpHw6ZObch).

We have updated our paper to make sure all reviewers’ concerns, if not articulated in the original submission, are fully addressed. Representatively:

- **Section 4.4, Appendix C, Appendix H:** Added evaluations on extensive RPT configuration variants to demonstrate that our findings are robust across **different RPT algorithms** (Reviewer 172Y), **different base models** (Reviewers 172Y, FGBZ, TWrh), **12 intermediate checkpoints** over **extended training steps** (Reviewers 172Y, TWrh), and **larger model sizes** (Reviewers 172Y, TWrh).

- **Appendix B:** Added quantitative analysis that effectively **reveals reasoning patterns across domains** (Reviewers FGBZ, 8H9o), further strengthening our discussions in Sections 2, 4.2, and 4.3.

- **Appendix D:** Added quantitative analyses of RPT **reward signals and training dynamics**, further validating the soundness of our study design and the reliability of our conclusions (Reviewer TWrh).

- **Appendix E:** Added **mixed-domain RPT** results, further strengthening our conclusion that domain isolation contributes to RPT's lack of generalizability (Reviewer TWrh).

- **Appendix F:** Added the full **per-model, per-task results** for all evaluations, further improving the transparency of our conclusions (Reviewer 172Y).

- **Appendix G:** Added **relative accuracy improvements** for all evaluations as a secondary metric, further supporting all claims in the main text (Reviewer 172Y).

We hope this summary is helpful in your final decision-making process, and are more than happy for further discussions.

Regards,

Paper 14011 Authors

---

### Meta-Review · Area_Chair_oZrq · 2025-12-16

**Summary:**

This paper asks a simple but very relevant question: when we do reinforcement post-training and see big gains, do those gains actually carry over to domains the model didn’t train on? Reviewers generally think the question matters, the paper is easy to follow, and the empirical setup is sensible. The main takeaway is pretty clear: the gains are often domain-specific, and transfer is much less reliable once you move beyond closely related reasoning domains. The weak spot is that some of the lack of transfer story can still be entangled with model/data differences and the particular RPT setup in the intervention experiments, so it’s hard to treat every result as fully causal. I’d want tighter claim wording, fuller per-domain/per-task breakdowns, and clearer documentation of training/eval details so readers can judge exactly what does (and doesn’t) transfer.

**Reviewer Concerns:**

**Reviewer 172Y**: Largely positive, but wanted cleaner evidence that the transfer limits aren’t an artifact of evaluation choices, and asked for clearer reporting / breakdowns to make the takeaways airtight.

**Reviewer TWrh**: Also supportive, but pushed on scope: more domains/tasks and more concrete analysis of what transfers (and why) would strengthen the paper beyond a descriptive result.

**Reviewer 8H9o**: Borderline, main concern is confounding in the observational comparisons (model family, training data, recipe differences), so they wanted tighter controls and clearer claim calibration.

**Reviewer FGBZ**: Most skeptical, felt the interventional study wasn’t broad enough (algorithms/backbones/variants) to support strong general conclusions, and wanted more systematic ablations to rule out alternative explanations.

**Reviewer Scores:**

Overall, the scores may shift upward somewhat, but the authors may not be able to fully resolve the negative reviewers’ concerns.

---

### Decision · Program_Chairs · 2026-01-26

Accept (Poster)